# Temperature-related neonatal deaths attributable to climate change in 29 low- and middle-income countries

Asya Dimitrova [1,2] ✉, Anna Dimitrova [3], Matthias Mengel [4], Antonio Gasparrini [5], Hermann Lotze-Campen [1,6] & Sabine Gabrysch [1,2,7]

Exposure to high and low ambient temperatures increases the risk of neonatal mortality, but the contribution of climate change to temperature-related neonatal deaths is unknown. We use Demographic and Health Survey (DHS) data (n = 40,073) from 29 low- and middle-income countries to estimate the temperature-related burden of neonatal deaths between 2001 and 2019 that is attributable to climate change. We find that across all countries, 4.3% of neonatal deaths were associated with non-optimal temperatures. Climate change was responsible for 32% (range: 19-79%) of heat-related neonatal deaths, while reducing the respective cold-related burden by 30% (range: 10-63%). Climate change has impacted temperature-related neonatal deaths in all study countries, with most pronounced climate-induced losses from increased heat and gains from decreased cold observed in countries in sub-Saharan Africa. Future increases in global mean temperatures are expected to exacerbate the heat-related burden, which calls for ambitious mitigation and adaptation measures to safeguard the health of newborns.

The steep increase in atmospheric concentrations of greenhouse gases (GHGs) from human activity since the start of the fossil age has led to increasingly apparent changes in the climate. The last decade has been the warmest on record, with global mean temperatures reaching 0.95 °C–1.2 °C above pre-industrial levels[1]. Exposure to high ambient temperatures poses a direct and immediate threat to human health, particularly for populations with low physiological and socio-economic adaptive capacities[2,3].

Exposure to extreme temperatures can be especially detrimental to the health of newborns because of their inherent physiological and anatomical vulnerabilities[4,5]. Neonates, i.e., infants in the first 28 days of life, especially those that are preterm and have low birth weight, have immature thermoregulatory systems and much narrower optimal body temperature ranges than adults[6]. Thermoregulation in neonates is also complicated by their higher metabolic rate (more energy expenditure) and lower sweating rate[7] and thus lower capacity to dissipate heat. In addition, neonates are characterized by smaller blood volume and higher heart rate, which impacts how their heart and blood vessels react to extreme temperatures[7,8]. Anatomically, neonates, particularly preterm neonates, have a high surface-area-to-mass ratio, which makes them prone to rapid heat loss and consequent hypothermia[9]. Conversely, this can also lead to higher heat absorption from the environment and an increased risk of dehydration and heat illness. Severe infections during the neonatal period, such as pneumonia and sepsis, can further exacerbate an infant's physiological susceptibility to heat and cold[10]. Ambient temperatures might affect

[1]Research Department 2, Potsdam Institute for Climate Impact Research (PIK), Member of the Leibniz Association, Potsdam, Germany. [2]Institute of Public Health, Charité – Universitätsmedizin Berlin, corporate member of Freie Universität Berlin and Humboldt-Universität zu Berlin, Berlin, Germany. [3]Scripps Institution of Oceanography, University of California, San Diego, CA 92037, USA. [4]Research Department 3, Potsdam Institute for Climate Impact Research (PIK), Member of the Leibniz Association, Potsdam, Germany. [5]Environment & Health Modelling (EHM) Lab, Department of Public Health Environments and Society, London School of Hygiene & Tropical Medicine, London, UK. [6]Department of Agricultural Economics, Faculty of Life Sciences, Humboldt University of Berlin, Berlin, Germany. [7]Heidelberg Institute of Global Health, Heidelberg University, Heidelberg, Germany. ✉e-mail: asya.dimitrova@pik-potsdam.de

very early neonates, i.e., infants in the first 24 h of life, through different physiological pathways compared to later neonates. Preterm birth and complications during childbirth are a leading cause of very early neonatal mortality in low- and middle-income countries (LMICs), while infections are more common among later neonatal deaths[11,12]. Non-optimal ambient temperatures, particularly heat, have been associated with preterm births and certain pregnancy complications that increase the risk of adverse perinatal outcomes, such as placental abruption, gestational hypertension, and gestational diabetes mellitus[13–15].

In 2019, there were an estimated 2.4 million neonatal deaths (95% CI: 2.3–2.7) globally, which accounted for nearly half (47%) of all under-5 child deaths[16]. The first week of life is especially critical, with 36% of neonatal deaths occurring on the day of birth and 73% occurring during the first week of life[17]. An overwhelming 91% of total neonatal deaths occur in LMICs, mainly in sub-Saharan Africa and South Asia[18,19]. Yet, most of the limited literature on ambient temperatures and neonatal mortality is based on populations living in temperate climates and high-income settings[20–24], with only two studies reporting effects for LMICs[25,26]. Furthermore, none of the existing studies on the link between temperature and neonatal mortality have attempted to quantify the impact of climate change. As exposure to high temperatures has become more common with climate change and given that neonates are particularly vulnerable to heat and cold, it is important to determine how much climate change has so far contributed to the burden of neonatal mortality.

Impact attribution studies use formal methods to evaluate the extent to which observed changes in natural and human systems can be attributed to recent climate change as opposed to other potential drivers[27]. Impact attribution studies can increase awareness of the already occurring impacts of climate change, inform society about related costs and damages, guide adaptation plans, and support climate-related litigation[28]. Most attribution studies to date have focused on observed impacts on natural systems[28], while impacts on human health, especially in children and for LMICs, have received limited attention[29,30].

Here, we address this gap by assessing the relative contribution of observed climate change to the burden of temperature-related neonatal deaths in 29 LMICs over the period 2001–2019 ($n = 40,073$). Building on the recent impact attribution framework of the Intersectoral Impact Model Intercomparison Project (ISIMIP)[31], we combined the largest internationally comparable dataset on neonatal deaths in LMICs[32] – the Demographic and Health Surveys (DHS) – with three recent temperature reanalysis datasets and corresponding counterfactual datasets in a two-stage analysis.

First, we conducted a time-stratified case-crossover analysis to quantify the non-linear association between daily ambient temperatures and neonatal mortality. We estimated the association for all neonatal deaths (0–28 days of age) as well as for very early neonatal deaths (0 days, i.e., <24 h of age) due to the potentially different exposure pathways. We standardized absolute temperatures in location-specific temperature percentiles to account for population adaptation to their predominant climate. We used conditional logistic regression and applied a distributed lag non-linear model to the pooled data across all countries to estimate the non-linear and delayed effects of temperature on the risk of neonatal mortality. Exposure-response associations were estimated for each of the three bias-adjusted global observational temperature datasets generated within ISIMIP. This allowed us to at least partly capture uncertainties from differences in temperature input data and reconstruction methods. Data on neonatal deaths were derived from DHS based on the reported age at death of each child. All newborn deaths that occurred within the first 28 days of life were classified as neonatal deaths, and those that occurred on the day of birth were classified as very early neonatal deaths. We included 40,073 neonatal deaths, 15,027 of which were very

early neonatal deaths. The dataset encompassed 29 countries, mainly in sub-Saharan Africa and Asia, spanning intersecting intervals between 2001 and 2019 (Supplementary Table 1).

Second, we used the estimated exposure-response functions to compute the burden of temperature-related neonatal deaths and very early neonatal deaths for each country under two scenarios – a factual scenario consisting of three historical temperature reanalysis datasets and a counterfactual scenario comprising the same three datasets but without the climate change signal. The counterfactual data share the climate and weather variability with the factual data but do not include the long-term warming trend due to global climate change[33]. We further calculated the temperature-attributable neonatal mortality rates using 2001–2019 country-specific data from UNICEF on total births and neonatal deaths[18]. We obtained the excess number of neonatal deaths attributable to climate change by subtracting the temperature-related burden in the counterfactual from those in the factual scenario. Following the IPCC Working Group II definition of impact attribution, we focus on the quantification of impacts from changes in climate, irrespective of the causes for the changes in climate[34]. The attribution of climate impacts to anthropogenic forcing would need an additional step separating anthropogenic climate forcing from other sources of climate trends.

## Results

### Epidemiological analysis

The changes in mean annual temperatures over time for each scenario and reanalysis dataset are depicted in Fig. 1a. Across all 29 locations and datasets, the annual mean temperature increased from 21.2 °C at the beginning of the 20th century to 22 °C in the first decade of the 21st century. During the study period (2001–2019), the annual mean temperature in the factual datasets was, on average, 0.9 °C higher than in the counterfactual. The patterns and magnitude of warming were similar for all three factual-counterfactual dataset pairs. There were, however, large differences between study locations in the observed level of warming during 2001–2019, ranging from -0.5 °C in Bangladesh to >1.2 °C in Senegal, South Africa, and Mali (Fig. 1b; see Supplementary Fig. 1 for country-specific time series).

The pooled non-linear exposure-response associations of ambient temperature (percentile) with overall and very early neonatal mortality for the three factual temperature datasets are presented in Fig. 2. Two of the factual datasets (20CRV3-W5E5 and GSWP3-W5E5) were identical for the time period of our study and are depicted together (Fig. 2b, d). The exposure-response associations represent the cumulative risk of mortality at every temperature percentile relative to the corresponding optimal temperatures over a 2-day lag period. For both overall and very early neonatal mortality, we observed a U-shaped exposure-response function. However, for the overall neonatal period, the risk of mortality was higher at lower temperatures (Fig. 2a, b), while for the very early neonatal period, the risk increased more steeply at higher temperatures (Fig. 2c, d). The very early neonates exhibited a lower optimal temperature, namely, at the 41.3rd or 40.3rd temperature percentile (depending on the factual dataset), compared to the overall neonatal population, for whom the optimal temperature was at the 51.9th or 53.6th temperature percentile. In absolute values, temperatures of minimum mortality across countries varied between 9 °C and 29 °C for neonates (see Supplementary Table 2 and Supplementary Fig. 2a) and between 5 °C and 28 °C for very early neonates (averaged between the three reanalysis datasets; see Supplementary Table 3 and Supplementary Fig. 2b). Higher optimal temperatures were observed in locations with higher mean annual temperatures and those closer to the equator (Supplementary Fig. 3).

In sensitivity analyses, restricting our sample to neonatal deaths reported within a shorter recall period prior to the interview produced very similar results (Supplementary Fig. 4). Using different knot placements also produced broadly similar results, but the number of

**a**

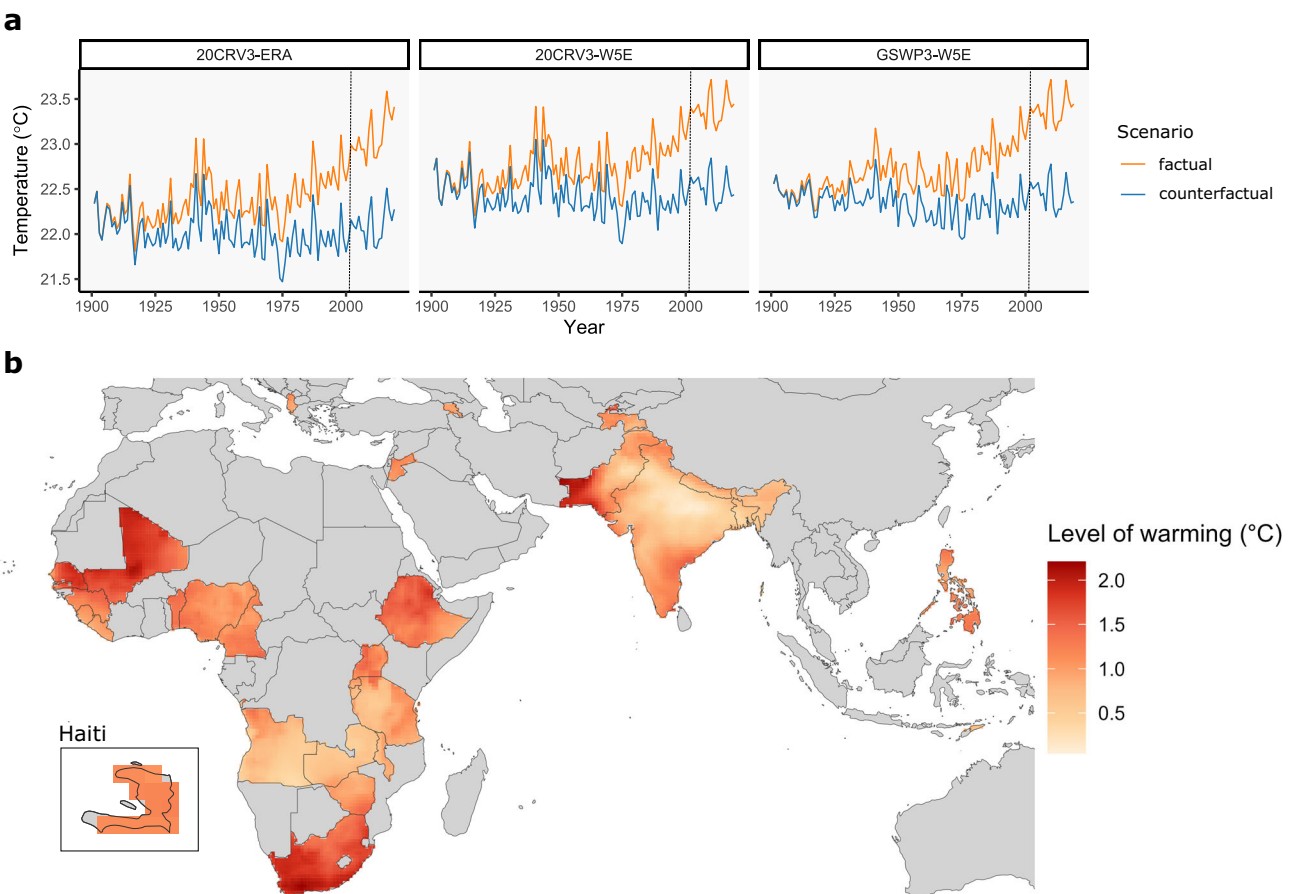

**Fig. 1 | Annual average temperature in factual and counterfactual scenarios. a** Annual average temperature across the 29 countries since 1900 by reanalysis dataset and scenario. **b** Average temperature difference between scenarios during 2001–2019 in the 29 study countries calculated as the mean across reanalysis datasets.

knots affected the shape of the exposure-response curve (Supplementary Fig. 5). We selected the model configuration that best-fit the data, as determined by the Akaike information criterion (Supplementary Table 4).

**Temperature-related neonatal deaths attributable to climate change**

Across all study locations, 4.3% (95% uncertainty interval (UI): 1.7–6.8%) of all neonatal deaths in the period 2001–2019 were associated with non-optimal temperatures in the factual scenario. Heat-related deaths accounted, on average, for 1.5% (95% UI: 0.2–2.6%) of the year-round burden of neonatal mortality across all included countries, and cold-related deaths accounted for 2.9% (95% UI: 1.5–4.1%) (See Supplementary Fig. 6 and Supplementary Table 5 for country-specific estimates). The four countries with the highest overall neonatal mortality rates, namely, Pakistan, Mali, Sierra Leone, and Nigeria recorded the highest temperature-related neonatal mortality rates (>160 neonatal deaths per 100,000 live births) (Fig. 3a, b). In terms of temperature ranges, moderately hot and moderately cold temperatures were responsible for the largest fraction of temperature-related neonatal deaths across all locations (Fig. 3b).

After subtracting the burden of temperature-related neonatal deaths in the counterfactual from the factual scenario, we estimated that 32% of the heat-related neonatal deaths (country range: 19–79%) in the period 2001–2019 can be attributed to climate change. In real terms, this amounts to 175,133 additional neonatal deaths (95% UI: 7806–353,516) (See Supplementary Table 6 for location-specific estimates). The heat-related neonatal deaths in the factual scenario represent a 46% increase compared to a counterfactual scenario

without climate change. The contribution of climate change to heat-related neonatal mortality was largest in the Philippines (79%), Haiti (79%) and Rwanda (70%) (Fig. 3f). In contrast, we find that in the period 2001–2019, climate change reduced the burden of cold-related neonatal deaths by an average of 30% (country range: 10–63%), equalling 457,384 (95% UI: 170,106–868,519) fewer neonatal deaths in total.

To account for differences in population size, Fig. 3c, d display changes in temperature-related neonatal deaths attributable to climate change as rates. These changes are estimated for the period 2001–2019 by comparing the factual scenario to the counterfactual scenario without climate change. Excess heat due to climate change has affected neonatal mortality rates across all study countries, with the largest increases in heat-related rates (>30 per 100,000) observed in Sierra Leone, Ethiopia, Liberia and Haiti (Fig. 3d). The largest positive effects (>110 per 100,000) from the reduction in cold-related neonatal mortality were observed in Liberia, Ethiopia, Sierra Leone, Uganda and Guinea (Fig. 3c). In terms of fraction of all neonatal deaths, the heat-related burden attributable to climate change ranged from 0.2% in Armenia to 1.1% in Haiti, while the averted burden from cold ranged from 0.3% in Albania, Nepal and Tajikistan to 4.6% in the Philippines (Supplementary Fig. 6 and Supplementary Table 5). Overall, the impacts of climate change on temperature-related neonatal mortality were largest in countries that had relatively high baseline neonatal mortality rates and at the same time experienced large temperature increases due to climate change (Sierra Leone, Ethiopia, Liberia, Mali, Guinea, Benin, Cameroon, Nigeria, Angola, Timor-Leste, Haiti).

For the case of very early neonatal deaths, in the factual scenario over the period 2001–2019, 4.1% (95% UI: 1.6–6.5%) were attributed to heat and 1.9% (95% UI: 0.2–3.5%) to cold, i.e., to temperatures above

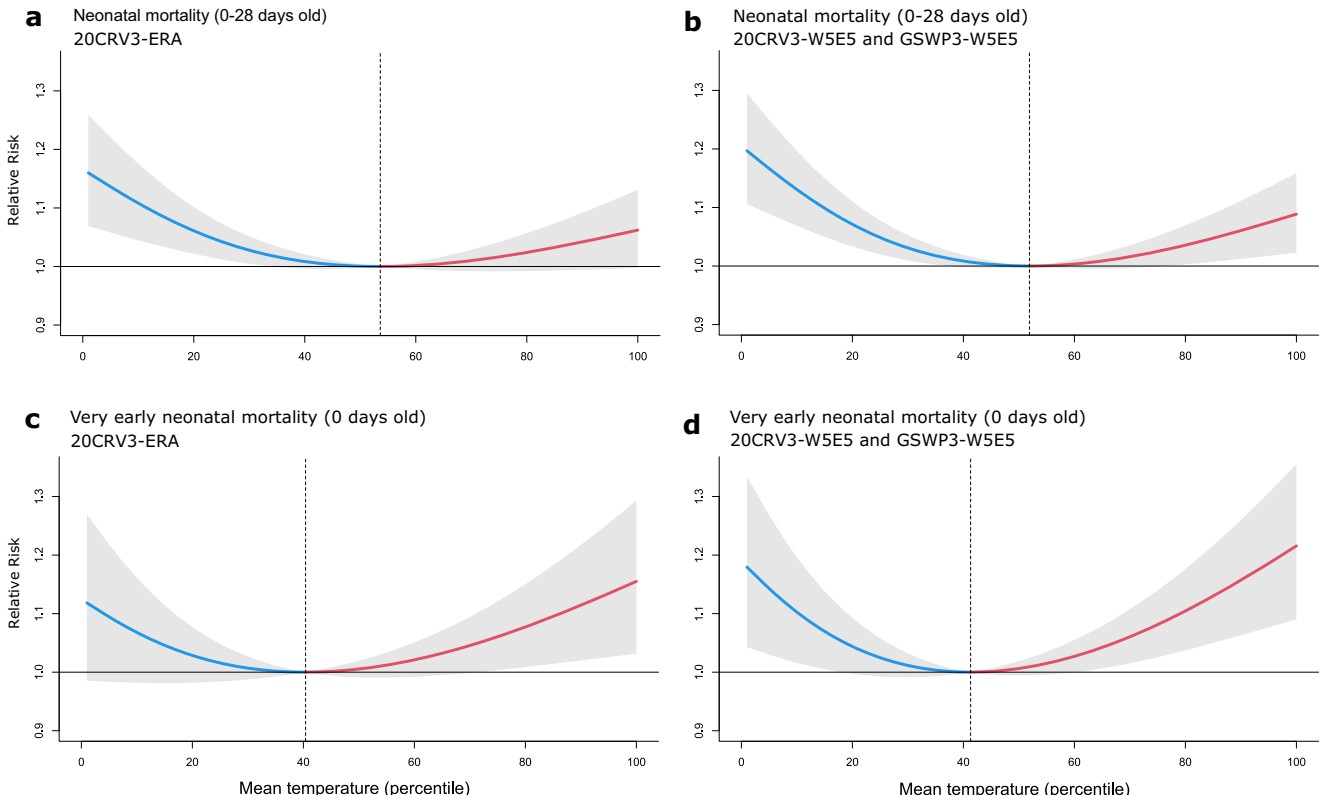

**Fig. 2 | Overall cumulative exposure-response associations for the three factual temperature datasets across the 29 countries. a** Association between temperature and neonatal mortality for global observational dataset 20CRV3-ERA5 and **b** datasets 20CRV3-W5E5 and GSWP3-W5E5. **c** Association between temperature and very early neonatal mortality for global observational dataset 20CRV3-ERA5 and **d** datasets 20CRV3-W5E5 and GSWP3-W5E5. Solid curves in blue (estimates below the optimum temperature) and red (estimates above the optimum

temperature) show pooled estimates of the exposure-response associations with 95% confidence intervals (shaded grey). Exposure-response associations are reported as relative risks for a cumulative 2-day lag of mean daily temperature (percentile) versus the optimum temperature (corresponding to the temperature of minimum mortality). Vertical dashed lines highlight the optimum temperatures. Data are presented as mean values with 95% confidence intervals.

and below the optimal temperature, respectively (See Supplementary Fig. 7 and Supplementary Table 7 for country-specific estimates). The countries where very early neonatal mortality was most prevalent also recorded some of the highest temperature-related mortality rates, namely, Liberia, Angola, and Timor-Leste (Fig. 4a, b). As with overall neonatal deaths, moderately hot and cold temperatures rather than the extremes dominated the temperature-related very early neonatal mortality burden (Fig. 4b).

Across countries, climate change contributed, on average, to 29% (country range: 8–72%) of the total burden of heat-related very early neonatal mortality, which in real terms equalled 168,835 very early neonatal deaths (95% UI: 48,835–296,467). Similar to overall neonatal deaths, the largest proportions (≥65%) of heat-related very early neonatal deaths attributed to climate change were observed in the Philippines, Haiti and Rwanda (Fig. 4f). We found that climate change reduced cold-related very early neonatal mortality by 35% (country range: 10–69%), amounting to 141,322 fewer neonatal deaths (95% UI: 2377–339,337) (see Fig. 4e and Supplementary Table 8 for country-specific estimates).

The largest increases in heat-related very early neonatal mortality rates induced by climate change (>32 per 100,000) were observed in Liberia, Timor-Leste, Sierra Leone, Ethiopia and Angola (Fig. 4d). Conversely, cold-related very early neonatal mortality rates averted due to climate change exceeded 37 per 100,000 live births in Liberia, Rwanda and Uganda, (Fig. 4c). In relation to all very early neonatal deaths in a country, the heat-related impact from climate change varied from null in Armenia to 3.2% in the Philippines. Concurrently, the cold-related fraction mitigated by climate change ranged from

0.2% in Albania, and Tajikistan to 3.7% in Uganda (Supplementary Fig. 7 and Supplementary Table 7).

## Discussion
Our findings indicate that both high and low ambient temperatures pose a risk to neonatal health in LMICs, with 4.3% of neonatal deaths during our study period attributable to non-optimal temperatures. This is consistent with existing evidence on the risk of neonatal hypothermia and hyperthermia in LMICs, even in tropical climates[10,35]. Previous findings on ambient temperatures and neonatal mortality risk are mixed, with some studies reporting associations with high temperatures only and others with low temperatures[14]. Similar to other studies[36], we find that moderately hot and cold temperatures dominate the temperature-related burden, which could be explained by their higher frequency throughout the year. We studied the first day of life separately to examine vulnerability to heat and cold, specifically during the very early neonatal period. We find that while neonates are overall more vulnerable to cold than heat, very early neonates seem to be at higher risk of mortality from heat-related causes. This may be explained by the different causal pathways through which ambient temperatures might affect neonatal mortality at different periods. Neonates who die within 24 h after birth are more likely to be infants born after complications during childbirth and prematurity[11,12]. Prematurity and other birth complications have been consistently associated with exposure to non-optimal ambient temperatures in utero, particularly heat[13–15]. Neonates who survive the first 24 h, on the other hand, are more likely to die from causes related to severe infections such as sepsis and

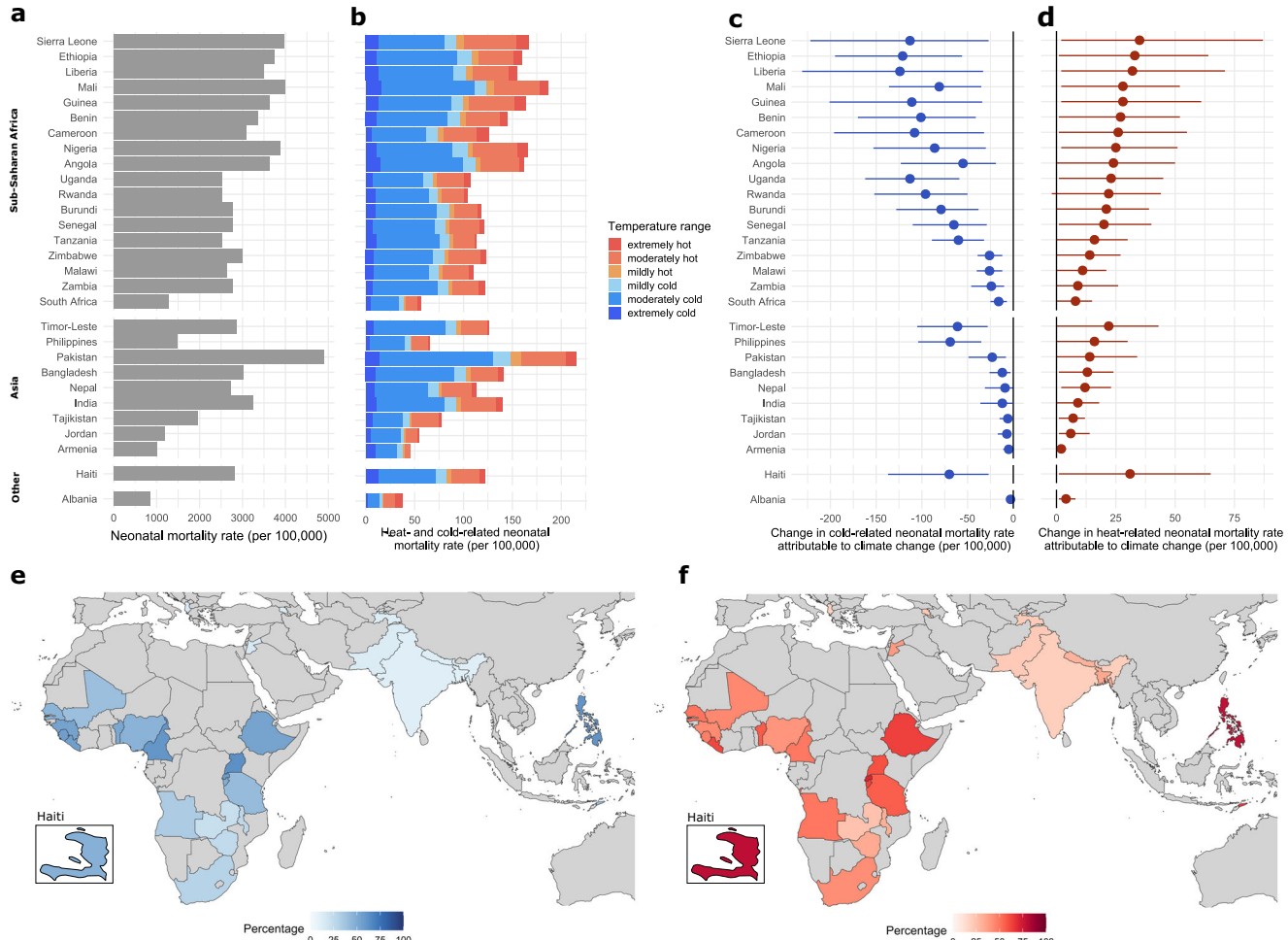

**Fig. 3 | Temperature-related neonatal mortality rates (0–28 days of age) and the contribution of climate change, 2001–2019. a** Neonatal mortality rate per 100,000 live births. **b** Temperature-related neonatal mortality rate per 100,000 live births by temperature range in the factual scenario. **c** Change in cold-related neonatal mortality rate per 100,000 live births attributed to climate change, with the corresponding 95% UI. **d** Change in heat-related neonatal mortality rate per 100,000 live births attributed to climate change, with the corresponding 95% UI. Uncertainty estimates in **c** and **d** were derived by generating 10,000 Monte Carlo simulations, assuming a multivariate normal distribution for the overall cumulative spline model coefficients for each of the three factual and counterfactual datasets. Point estimates are calculated as the average of the difference between the factual and counterfactual across all sample coefficients for the three temperature datasets. **e** Proportion of cold-related neonatal mortality averted due to climate change (% of cold-related neonatal mortality) **f** Proportion of heat-related neonatal mortality attributed to climate change (% of heat-related neonatal mortality).

pneumonia[11,12], which have generally been associated with hypothermia[6,37].

Our results demonstrate that by altering ambient temperatures, climate change is already directly affecting the survival of newborns in LMICs across the world. The upward shift in temperatures observed over the last two decades due to climate change has substantially increased the number of neonatal deaths through increasing heat-related mortality while concurrently diminishing the toll related to cold temperatures. Climate change impacts have been uneven, with the largest losses from increased heat but also gains from decreased cold observed in countries that had high baseline neonatal mortality rates and experienced the most warming. While the decline in the cold-related burden may aid efforts to reduce neonatal deaths, with ongoing climate change, these gains are likely to be outstripped by progressively increasing heat-related mortality in the future[38,39]. Health impact projection studies are warranted to fully understand the future health risks to pregnant women and newborns posed by climate change, especially in LMICs with limited adaptive capacities, frequent neonatal deaths, and large exposed populations.

Since we investigate both heat and cold components, use temperature percentiles, and focus on understudied population groups,

our results are not directly comparable to other climate impact attribution studies but are in line with existing evidence on substantial heat-related mortality burdens in adults from climate change[40–42]. Only one attribution study to date has attempted to quantify the health burden of climate change among children. Chapman et al. (2022)[43] found that climate change doubled heat-related child mortality in Africa in the period 2009–2020, which is higher than our estimate of a 46% increase in heat-related neonatal mortality. In addition to the wider age group of up to 5 years, unlike our analysis, the above study did not use population-specific exposure-response functions but applied published estimates of the relationship between ambient temperature and mortality from a few selected populations in Africa to the whole continent.

Our findings highlight the need for public health interventions to protect newborns in LMICs from low and high ambient temperatures. Prevention of postnatal thermal loss in low-resource settings is a recognized challenge, which has been linked to inadequate thermal control practices (i.e., insufficient heating of the birthplace, placing of the uncovered newborn on the ground or other cold surfaces, delayed wrapping, early bathing), insufficient knowledge of hypothermia diagnosis, lack of robust and affordable incubators and other infant

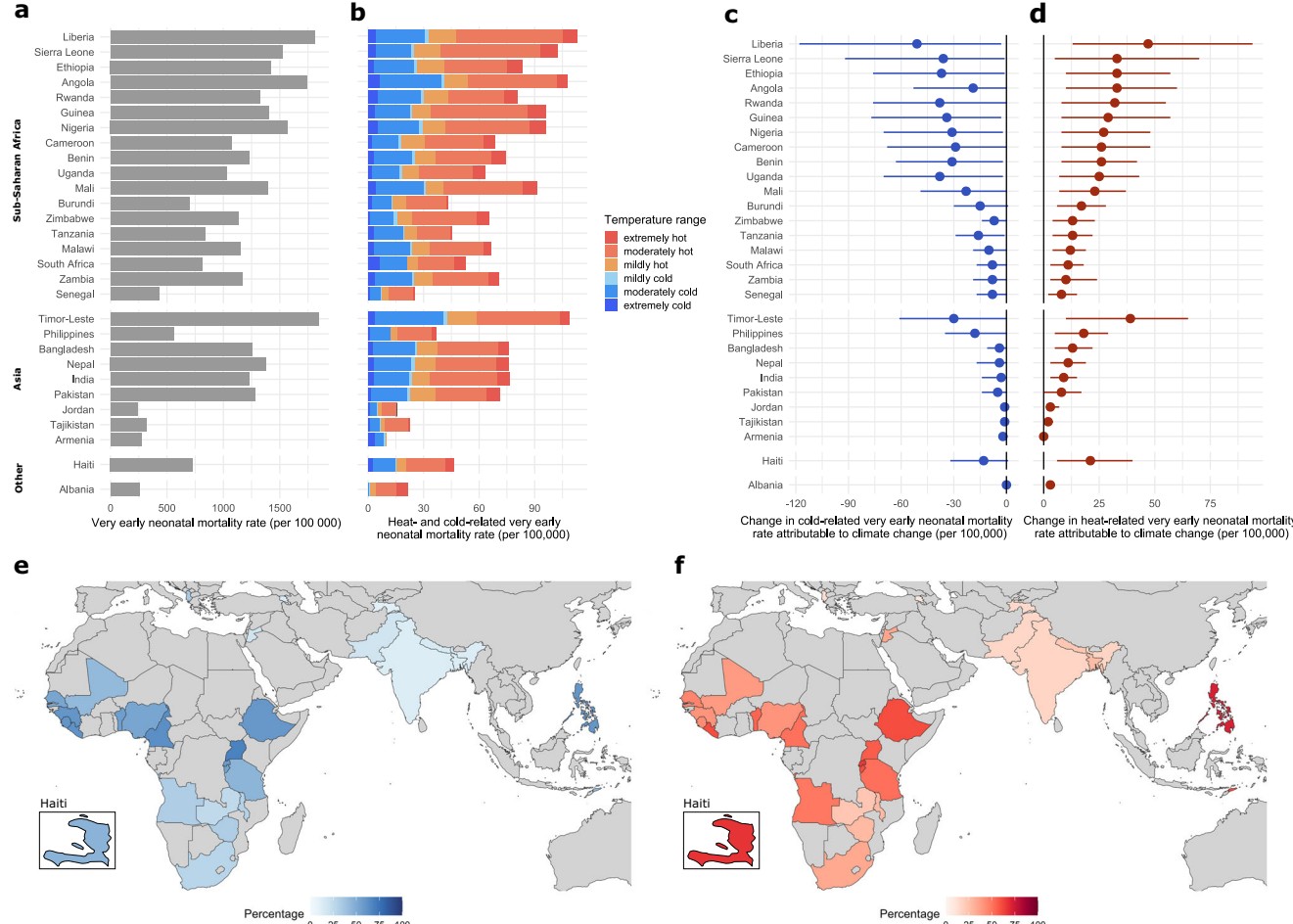

**Fig. 4 | Temperature-related very early neonatal mortality rates (0–1 days of age) and the contribution of climate change, 2001–2019. a** Very early neonatal mortality rate per 100,000 live births. **b** Temperature-related very early neonatal mortality rate per 100,000 live births by temperature range in the factual scenario; **c** Change in cold-related very early neonatal mortality rate per 100,000 live births attributed to climate change, with the corresponding 95% UI. **d** Change in heat-related very early neonatal mortality rate per 100,000 live births attributed to climate change, with the corresponding 95% UI. Uncertainty estimates in **c** and **d** were derived by generating 10,000 Monte Carlo simulations, assuming a multi-variate normal distribution for the overall cumulative spline model coefficients for each of the three factual and counterfactual datasets. Point estimates are calculated as the average of the difference between the factual and counterfactual across all sample coefficients for the three temperature datasets. **e** Proportion of cold-related very early neonatal mortality averted due to climate change (% of cold-related very early neonatal mortality) **f** Proportion of heat-related very early neonatal mortality attributed to climate change (% of heat-related very early neonatal mortality).

medical equipment and unreliable power supply[44]. On the other hand, the risk of heat stress in neonates can be exacerbated by poor post-natal practices (placing newborns in direct sunlight without shading, over-bundling in clothes and blankets), insufficiently frequent breast-feeding in hot weather leading to dehydration, failure of essential infant equipment and power outages in medical facilities during heatwaves[45]. Effective behavioural and community interventions to reduce mortality risks from neonatal hypothermia and hyperthermia include educational programs among mothers and other caregivers and training of health providers on practical measures that ensure thermal stability (e.g., immediate drying, skin-to-skin contact, delayed bathing)[45–47]. The provision of essential infant devices such as incubators and infant warmers tailored to the context in LMICs are important for preventing cold stress[44] but can be prohibitively expensive. Initiation of kangaroo mother care immediately after birth is an alternative low-cost warming method, which has proven effective in reducing neonatal hypothermia, severe infections, and mortality[48,49]. Apart from good postnatal thermal practices, improving the designs of homes and health facilities, increasing urban green spaces, and providing support for home cooling are some strategies that can help reduce heat stress for pregnant women and newborns[45]. It is important to note that these actions should be part of a broader effort to reduce neonatal deaths, given that temperature-related deaths constitute less than 5% of neonatal mortality according to our findings.

Some limitations and assumptions in our analysis need to be acknowledged. DHS data are based on self-reported birth history, and we included information on neonatal deaths within 15 years of the interview date. Furthermore, DHS data may not fully capture neonatal deaths due to social stigma in reporting, omissions, and mis-recording by the interviewer or misclassification between very early neonatal deaths and stillbirths[32]. However, since case-crossover is a self-matched design, meaning each case is compared to itself, and not others and misreporting of deaths (including inaccurate date due to too long recall periods) is likely to be non-differential and not dependent on daily temperature, omission and misreporting of neonatal deaths is unlikely to have affected our results. We were able to pool observations from 29 countries with distinct climates by characterizing the temperature exposure in percentiles. Thus, we could partly account for physiological and behavioural population adaptation to their local climate by allowing for location-specific minimum mortality temperatures. A main limitation of this approach is that it did not allow a flexible change in the shape of the exposure-response

function across locations. The use of temperature percentiles might have also led to underestimation of the impact of extreme heat events as the difference between extreme temperatures and moderate temperatures may be narrowed with this method. We also did not consider changes in the exposure-response function over time that may result from adaptation. However, such changes are likely to be predominantly influenced by factors not driven by climate, such as socio-economic development, improved health care services or infrastructural development, which evolve simultaneously but independently from the changing climate. Furthermore, the generalizability of our findings is weakened by the disproportionate share of India in our sample. Also, certain regions, such as Latin America, are largely underrepresented in our analysis due to the lack of recent DHS rounds with detailed day of birth and death information there. Routinely collected data on neonatal mortality from vital registrations or other reliable data sources with geographic coordinates would be needed to model location-specific exposure-response functions more flexibly and provide more generalizable estimates. Finally, our analysis did not distinguish between impacts induced by anthropogenic greenhouse gases and other influences on climate trends, which would need a more complex study design, including model ensembles[33].

This study provides the first evidence of the contribution of historical climate change to temperature-related neonatal deaths. We derived an exposure-response association applying state-of-the-art epidemiological methods to the largest internationally comparable dataset on neonatal deaths in LMICs and the most recent temperature reanalysis datasets. To fully understand the shifting balance between heat- and cold-related burdens, we investigated the continuum of all temperature-related neonatal deaths. The data and methodology used allowed us to report results at the national level and account for uncertainty stemming from the epidemiological estimates and the baseline temperature time series. The presented evidence demonstrates that climate change has already affected child health, both by exacerbating heat-related and alleviating cold-related neonatal deaths. With current policies in place, global average surface temperature is projected to increase by the end of the century to 2.8 °C above pre-industrial levels[50] compared to the current 1.1 °C, which is likely to lead to progressively increasing heat-related burdens and diminishing cold-related gains. Our findings add to the accumulating evidence of the large burden of climate change on the health of younger generations and underscore the need for ambitious mitigation and effective adaptation measures to safeguard the health of the most vulnerable populations.

## Methods

### Data on neonatal deaths
We derived data on neonatal deaths from Demographic and Health Surveys (DHS). DHS are large nationally representative, cross-sectional household surveys that provide comparable data across countries for a wide range of demographic and health indicators, including reproductive and child health. Households are selected randomly through a two-stage probability sampling procedure based on an existing sampling frame. All women of reproductive age (15–49 years) from the selected households are interviewed by trained fieldwork staff, including on their birth histories. Since its launch in 1984, the DHS program has conducted surveys in more than 90 low- and middle-income countries (LMICs), systematically collecting data on neonatal deaths, and is thus the largest publicly available source for such data[32]. We restricted our analysis to surveys with available (i) global positioning system (GPS) data, i.e., latitude and longitude, of primary sampling units (PSUs); (ii) detailed information on the date of birth and death (day, month, and year) and (iii) temperature data for at least 2 of the 3 ISIMIP datasets. In urban areas, a PSU could represent a neighbourhood or a cluster of city blocks; in rural areas, it could represent a village or a group of villages. To ensure respondent confidentiality,

GPS coordinates of PSUs are randomly displaced by up to 2 km in urban areas and 5 km in rural areas, with an additional 1% of PSUs displaced by up to 10 km. We included in our analysis only surveys from DHS-VII and DHS-VIII, collected between 2014 and 2021, which include information on the exact date of birth and death compared to earlier surveys, which only recorded the month and year of birth and death. This allowed us to determine a 2-day period prior to the day of birth as our exposure window. The final analysis included data from 32 DHS covering 29 countries (Supplementary Fig. 8). The data are publicly available for academic research and can be accessed on the DHS website (www.dhsprogram.com) upon prior registration.

We retrieved data on neonatal deaths based on the age of death of live-born babies. All newborn deaths that occurred within the first 28 days of life were classified as neonatal deaths, and those that occurred on the day of delivery were classified as very early neonatal deaths. We restricted the neonatal deaths to those reported by the mother within 15 years prior to the interview date.

### Temperature factual and counterfactual datasets
We based our analysis on three surface temperature datasets that are part of the ISIMIP3a simulation round, namely, 20CRV3-ERA5, 20CRV3-W5E5, and GSWP3-W5E5[51]. We discarded the fourth ISIMIP3a dataset 20CRV3, as it ends in 2015 and does not have full overlap with the study period (2001–2019). The datasets combine the latest-generation reanalysis data from 1979 to today (ERA5 and W5E5)[52,53] with reanalysis data that capture the earlier period from 1900 to 1978 (20CRV3 and GSWP3)[54,55]. Reanalysis datasets combine a wide range of temperature observations, including from weather stations, aircraft, ships, satellites, and other sources, along with weather forecasting models to generate complete and consistent temperature time series within high-resolution grids, including for world regions where observational temperature data are missing or sparse. The 1900–1978 part is homogenized with the data from ERA5 or W5E5 through bias adjustment[56]. The combined datasets start in a period when climate change was largely absent and thus allow the construction of counterfactual versions from the dataset without the use of climate model data. The two factual W5E5-based datasets are identical during the period of the study data but differ in their counterfactual that is influenced by the data before 1978. The counterfactual datasets describe ambient temperatures in the absence of climate change. They were generated from the factual datasets by using an innovative detrending method to remove long-term variations linked to global climate change[33]. The method maintains the internal variability of the observed data by ensuring that temperature observations in the factual and counterfactual for a particular day have the same rank in their respective statistical distributions, which preserves the timing of climatic events[33]. The detrending is not sensitive to the cause of historical climate trends (e.g., anthropogenic emissions, external natural forcing or internal variability), following the impact attribution definition of the IPCC AR6[34]. Both the factual and counterfactual temperature datasets are gridded at a 0.5° × 0.5° (~55 km × 55 km) spatial resolution.

Since we combined data from populations spanning different climate zones, we converted the absolute temperatures into PSU-specific temperature percentiles, following previously described methods[57]. This approach allowed us to account for population adaptation to their predominant climate. We linked the individual cases of neonatal death with the daily mean ambient temperature data using the geographic coordinates of each PSU.

### First stage model: epidemiological analysis
We used a time-stratified case-crossover design to quantify the association between ambient mean temperature exposure, expressed as a cluster-specific temperature percentile, and the risk of neonatal death. The case-crossover design is well established in the literature on the acute health effects of short-term environmental exposures and has

been extensively applied in the study of the association between ambient temperatures and adverse birth outcomes[14,15,58,59]. In this study design each case serves as his/her own control, whereby exposure of the same individual on a case day is assessed against referent exposures on days before or after the case day. The main advantage of this approach is that observed and unobserved time-invariant individual confounders such as age, education, health care access, socio-economic status, and other factors are controlled for by design[60]. Following the time-stratified strategy, we matched each case day with control days, which were on the same day of the week within the same month and year as the neonatal death. Temperature conditions on the case day were compared with three or four control days. Restricting controls to the same day of the week addresses potential week-varying confounders while selecting controls within the same month and year accounts for long-term trends and seasonality. We applied conditional logistic regression on the pooled neonatal mortality data across all PSUs and countries to estimate the relative risks (RRs) of neonatal deaths at each temperature percentile compared to the optimum temperature, corresponding to the temperature percentile of minimum risk for neonatal mortality. We did not include air pollution in the model because daily data are unavailable. However, the existing literature indicates modest[20,61,62] or no[63-65] confounding effect of air pollution on the temperature-mortality or -preterm birth associations. We also did not control for relative humidity in the analyses due to minimal observed confounding effects in previous studies[66,67]. The non-linear and delayed exposure-lag-response relationships between temperature and neonatal deaths were modelled using distributed lag non-linear models (DLNMs). As per DLNM methodology[68], we used two spline functions defined within a cross-basis term to model the bi-dimensional exposure-lag association. We included a lag period of up to two days before the event to examine the delayed effects of low and high ambient temperatures on neonatal deaths. The modelling choice of the lag-neonatal mortality associations was tested with lags up to seven days prior to the event (Supplementary Figs. 9 and 10). We used the Akaike Information Criterion (AIC) model selection to assess a set of possible models with different knot placements for the temperature-neonatal mortality associations (Supplementary Table 4). The best-fit models included a natural spline function with one knot at the 20th percentile of the temperature distribution for the association between temperature and neonatal mortality and at the 10th percentile for the association between temperature and very early neonatal mortality and natural spline functions with one internal knot at equally spaced log-values over two lagged days for the lag-response dimensions. To determine the temperature of minimum mortality, we generated predictions of the RR on a matrix of values of the original predictor (temperature percentiles) and lags using the dlnm package. We then selected as minimum mortality or optimal temperature the temperature percentile value where the predicted RR was minimised. Temperature percentiles below and above that value were considered non-optimal.

We performed several sensitivity analyses to evaluate the robustness and strength of the temperature-neonatal mortality association. First, we tested different configurations of the exposure-response function such as the number and placement of knots (See Supplementary Table 4 for more details). Second, to reduce any potential effect of misreporting or inaccurate reporting of deaths on our results due to problems with recall, we re-estimated the association after restricting the sample to deaths reported within 5 and 10 years prior to the interview.

**Second-stage model: impact attribution**
In the second stage, we used a backward approach to calculate the attributable fraction (AF) of neonatal deaths associated with non-optimum temperature under both the factual and counterfactual scenarios for the period 2001–2019[69]. This method allows to incorporate the added temporal dimension in exposure-response relationships with complex temporal patterns and has been extensively applied in past research using the DLNM framework[70,71].

More specifically, for each case day of the sample the attributable fraction $AF_{x,t}$ was calculated using the following formula:

$$AF_{x,t} = 1 - e^{\left(-\sum_{l=l_0}^{L}\beta_{x_{t-l}},l\right)} \tag{1}$$

where the parameter $\beta_x$ represents the risk associated with exposure $x$ relative to the reference value of $x_0$ and corresponds to logarithm of the odds ratio; $l_0$ and $L$ represent the lowest and the highest number of lags considered, respectively; and $\sum_{l=l_0}^{L}\beta_{x_{t-l}},l$ represents the overall cumulative exposure-response association, which is composed of the sum of contributions $\beta_{x_t}$ from exposures $x_{t-l_0},\dots,x_{t-L}$ experienced within the lag period. Thus, $AF_{x,t}$ can be interpreted as the fraction of neonatal mortality at time $t$ attributable to past temperature exposures in the period $t - l_0,\dots t - L$ compared to a constant exposure $x_0$ throughout the same period. These attributable risk fractions can be interpreted as individual's probability of dying and summed up to derive attributable fraction at the sampled population level. We separated the attributable fractions into components related to hot and cold temperatures by restricting the analysis to temperature ranges above and below the reference temperature percentiles for neonatal mortality. For the factual scenario, we additionally calculated the attributable fractions for five temperature ranges: extremely cold, moderately cold, mildly cold, mildly hot, moderately hot, and extremely hot, with cut-offs corresponding to the 97.5th, 75th, minimum mortality temperature, 25th and 2.5th temperature percentiles. The attributable fractions were calculated for each dataset, country, and scenario. We subsequently obtained the excess fraction of neonatal deaths attributable to climate change by subtracting the hot- and cold-related burdens between the factual and counterfactual scenarios. Due to disparity in the historical temperature series (factual scenario) for two of the three reanalysis products, we estimated the exposure-response associations and the respective temperature-related mortality burdens separately for each dataset (Supplementary Figs. 11 and 12). We computed the climate change-attributable heat- and cold-related neonatal mortality rates per 100,000 live births for each country as the climate attributable fractions multiplied by the respective neonatal mortality rates in the period 2001–2019 retrieved from UNICEF[18] (see Supplementary Table 9).

$$AR_{neonat\ mortality,} = m_{neonat} \cdot AF_{CC} \tag{2}$$

where $m_{neonat}$ refers to the neonatal mortality rate and $AF_{CC}$ – to the fraction of heat- or cold-related neonatal mortality attributable to climate change. We derived very early neonatal mortality rates for the same period by first estimating the share of overall neonatal deaths for each country using the DHS data and applying this share to the country-specific neonatal mortality statistics from UNICEF. The comparative risk assessment framework that we used did not technically allow us to apply the DHS sampling weights in the estimation of the mortality burdens, but this should not have considerably affected the representativeness of the estimated attributable risk fractions. We quantified the uncertainty in the attributable risk fractions related both to the parameters of the exposure-response functions and the variation in temperature series across the three datasets. Specifically, we generated 10,000 samples of the regression coefficients through Monte Carlo simulation, assuming a multivariate normal distribution for the overall cumulative spline model coefficients for each of the three factual and counterfactual datasets. Point estimates were calculated as the average of the difference between the factual and counterfactual across all sample coefficients for the three temperature datasets. The lower and upper bounds of the empirical confidence intervals were estimated as the 2.5th and 97.5th percentiles

of the empirical distribution across coefficient samples. We performed all analyses in R (version 3.6.1) using the packages *dlnm* and *survival*.

## Reporting summary

Further information on research design is available in the Nature Portfolio Reporting Summary linked to this article.

## Data availability

The survey data used in this study are publicly available from the Demographic and Health Surveys (DHS) website upon prior registration (https://dhsprogram.com/). We included all surveys that contained birth history information (Births Recode files), detailed information on the date of birth and death (day, month, and year), and global positioning system (GPS) coordinates of the primary sampling units (Geospatial Covariates files). A complete list of surveys included in the analysis can be found in Supplementary Table 1. Country-specific data on total births and neonatal deaths were retrieved from UNICEF. The factual and counterfactual temperature time series used in the analysis are publicly available from the repository of the Inter-Sectoral Impact Model Intercomparison Project (ISIMIP) (https://data.isimip.org/). We based our analysis on three surface temperature datasets that are part of the ISIMIP3a simulation round: 20CRV3-ERA5, 20CRV3-W5E5, and GSWP3-W5E551.

## Code availability

The codes generated in this study have been deposited in the following GitHub repository: https://github.com/DimitrovaAsya/attri_temp_neonat_mort.

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

## Acknowledgements

The authors are thankful to Dr. Nicholas Kyei for reviewing and providing feedback on the final manuscript. S.G. received funding through a Recruiting Grant from Stiftung Charité (BIH_PRO_540). Part of this research has received funding from the German Federal Ministry of Education and Research (BMBF) under the research project QUIDIC (01LP1907A) and is based upon work from COST Action CA19139

PROCLIAS (PROcess-based models for CLimate Impact Attribution across Sectors), supported by COST (European Cooperation in Science and Technology; https://www.cost.eu).

## Author contributions

As.D., An.D., S.G., H.L. conceived and designed the study. As.D. and An.D. retrieved, pre-processed and validated the data, supported by S.G. As.D., An.D., A.G. and S.G. contributed to model development and/or revisions. As.D. carried out the analysis. As.D. wrote the first draft of the manuscript. An.D., S.G., A.G., M.M. and H.L. contributed to subsequent versions, and to the interpretation of the data and results. All authors approved the final version of the manuscript.

## Funding

## Competing interests

The authors declare no competing interests.
