## [Peer Review File · Nature Communications]

Temperature-related neonatal deaths attributable to climate change in 29 low- and middle-income countriesREVIEWER COMMENTS

Reviewer #1 (Remarks to the Author):

This study uses time-stratified case-crossover analysis to quantify the non-linear association between daily ambient temperatures and neonatal mortality and then estimates the attributable fraction of neonatal mortality attributed to non-optimal temperature. In addition, the authors also calculated the excess number of neonatal deaths attributable to climate change. These findings could provide valuable evidence for the impact of climate change on neonatal mortality. However, there are still some concerns that should be addressed by the authors.

Major comments

1. Line 295. It is hard to say that the result derived from this study is global evidence since samples are collected from limited countries mainly in Africa. More importantly, almost half of the deaths were located in India, which weakens the representativeness of the main findings.
2. Line 336. What is the distribution of the time gap between the occurrence of neonatal deaths and the interview of the mothers? A maximum of 15 years is too long to recall the accurate time of neonatal deaths, which may cause substantial recall bias.
3. Line 361. Indeed, the temperature percentiles may to some extent account for the population adaptation to their local climate. However, this method may lose considerable statistical power, since plenty of information on the temperature was omitted. For example, one of the main impacts of climate change is the substantial increase in extreme heat events, especially in warm regions. However, when using temperature percentiles, the difference between extreme temperatures and moderate temperatures may be narrowed.
4. Line 378. More detailed information about the model specification should be presented. For example, Are any confounding factors, such as air pollution and humidity, adjusted for?
5. The authors combined data from all countries into one model, which implied a fixed effect of temperature on neonatal deaths for all countries. This assumption may be not appropriate, especially when considering that almost half of the samples were from one country (namely India). Plenty of previous epidemiology studies confirmed that the association between temperature and mortality varied by region due to different climate conditions, socioeconomic factors, and so on. Thus, in line with previous studies, I would suggest a two-stage analysis strategy, namely modeling the impact of temperature for each country and then pooling the effects of all countries by random effect meta method. Since many countries have limited sample sizes, the BLUP method could be applied to re-estimate the country-specific effect by borrowing information from other countries.
6. Sensitive analyses are needed to verify the robustness of the main analyses, especially the exposure-response association between temperature and death.
7. Line 386. Why authors choose the lag period of up to two days as the maximum lag period of temperature? Previous studies indicated that the impact of cold on mortality may persist for up to weeks. Authors can provide more information about the lagged effect of temperature, especially the difference between two kinds of death since their underlying mechanisms may be different. In addition, it would provide valuable information for the prevention of neonatal death.
8. Line 406. Is it the "cut-offs corresponding to the 97.5th, 75.5th, minimum mortality temperature, 25.5th, and 2.5th temperature percentiles"?
9. The authors calculate the excess fraction of neonatal deaths attributable to climate change by simply subtracting the hot- and cold-related burdens between the factual and counterfactual scenarios. This calculation completely ignores human adaptation to temperature changes. As the authors indicated in the manuscript, people can adapt to local climate conditions, so they may also adapt to local climate changes. Thus, the excess fraction of neonatal deaths attributable to climate change may be overestimated (for heat).

Reviewer #2 (Remarks to the Author):

General comment:

This is very well written paper with appropriate methodology applied that links the important topic of climate change and neonatal mortality burden in low- and middle-income countries.

Comments

- It would greatly benefit the readers if the authors explained in detail how 'non-optimal' or minimum mortality temperature percentile was calculated.
- I wonder if the authors could access data from the health and demographic surveillance sites in LMIC implementing verbal autopsy to corroborate the findings?
- Why was the data pooled yet country-specific analysis could be done?
 - o Would the results differ if you did country-specific analysis for the exposure response curves and then combined the results later in a meta regression?
 - o Would it matter to account for country specific factors that influence neonatal outcomes in a multilevel kind of analysis?
- You mention, community specific percentiles, what are the communities in this respect?
- What would be the best mitigation strategies to reduce the burden?
- I wonder how this study based on the design accounts for other competing risk factors in these communities such as high malaria burden, health systems etc
- So many countries are not represented, could this tilt the outcome if there were more countries with DHS data?
- Can the attribution be extended to the future e.g 2030s ,2050s based on projections data?

Reviewer #3 (Remarks to the Author):

Temperature-related neonatal deaths attributable to climate change

This is a very interesting study and the findings are meaningful and useful. The results fill in a gap in current knowledge about temperature effects on neonates, where empirical data is being used to calculate the climate attributable burden of neonatal mortality. Presenting the % change in deaths from heat and cold and associated number of deaths provides policy relevant findings. It's great to see countries in Africa and Asia are predominantly featured, where evidence is lacking – and the country specific effect estimates are useful for policymaking/guiding uptake of local adaptation measures. I thank the authors for publishing this work.

My concern is that there is an uneven mix of countries making up the 'meta-estimate' grouping when I stratified the 29 countries across continents (Africa, Asia, Caribbean, Europe), i.e., Europe has one country, while Africa has 18. How was this taken into account in the meta-analysis, as the bias in sampling countries may place limitations on comparing effects across continents for the meta-estimates? Can you please explain your approach and how you dealt with the heterogeneity in terms of providing representative meta-estimates. The study also misses large LMIC demographics, for example, in Latin America. Can you please justify this? This is a limitation and should be talked about in the discussion.

Furthermore, a country like Jordan is strictly speaking in the middle-east, however, this is not a grouping presented in this study (I understand why you may have grouped it into Asia, where you have more countries). How were the groupings created? Rather than arbitrarily assigning groupings by continent, grouping by a climate classification such as the Koppen Geiger classification can enable effect estimates from countries with more homogeneous climates to be compared and meta-analysed. Can you please discuss your use of grouping by continent, and why this was selected (and what the pros and cons are)?

I can see in line 321 you give the rationale for including 29 out of 90 countries collecting DHS data as "We restricted our analysis to surveys for which global positioning system (GPS) data, i.e., latitude and longitude, of primary sampling units (PSUs) were available". Was this the only rationale, or was it the availability/completeness of the data?

A general comment on style: I find active voice to be more effective and engaging for the reader – it would be nice if the authors can consider using this in future publications.

For example line 225: On the other hand, climate change was estimated to have reduced cold-related very early neonatal mortality by 35% (range: 10-69%), amounting to 141,322 fewer neonatal deaths (95% UI: 2,377-339,337) (see Fig. 4e and Supplementary Table 3 for country-specific estimates).

Could be rephrased as: We found that climate change reduced cold-related very early neonatal mortality by 35% (range: 10-69%), amounting to 141,322 fewer neonatal deaths (95% UI: 2,377-339,337) (see Fig. 4e and Supplementary Table 3 for country-specific estimates).

Please find my specific comments attached below:

Title and abstract

Title: In the title please include: (i) sample of neonatal deaths analysed (40,073); (ii) a statement such as 'an analysis of 29 low- and middle-income countries' to indicate the scope.

Line 11: 29 low- and middle-income countries

Please add the following text before 29 so the sentence reads: Demographic and Health Surveys (DHS) data from 29 low- and middle-income countries.

Line 15 between "30%. [insert here.]Countries" please insert effect estimates on the range of mortality burden attributable to temperature between individual countries you included (i.e., the individual country that had the highest and lowest burden)?

Line 15: Countries in sub-Saharan Africa have experienced the most pronounced climate-induced losses from increased heat and gains from decreased cold

In the results component of the abstract there is no indication of effect estimates (% deaths) presented for region groupings other than Africa (other than 'all countries'). This could be interpreted as the authors only reporting positive or significant results in the abstract. For Africa, please describe the % of deaths attributable for heat and cold. Please also describe effect estimates (% deaths) from meta-analysis of other regional groupings. Furthermore, please clearly describe how the 29 countries were grouped for meta-analysis (was it by continent, by Koppen Geiger classification?). A short sentence on this would add clarity (I see from further reading that it's by continent).

Please conduct the meta estimates of the associations. Please include meta estimates in figures 3 and 4 with the extended figures 2, 3 4 and 5.

Line 19: the most vulnerable populations.

Considering your population of interest are neonates, please be specific and use 'neonates' here, rather than speaking generally about vulnerable populations.

Introduction

Lines 36 to 42: please use consistent terminology - neonates or infants (not interchanging use both):

Line 71: increase awareness of what?

Line 72: inform societal responsibility for the cost of damages and losses. I would change this sentence to "inform society about costs and damages due to climate change"

Line 79-81: we combined the largest internationally comparable dataset on neonatal deaths in LMICs with three recent temperature reanalysis datasets and corresponding counterfactual datasets in a two-stage analysis.

Please state what the empirical 'largest internationally comparable dataset' is.

Line 77: to the burden of temperature-related neonatal deaths in 29 LMICs over the period Please state the total sample size of deaths analysed in the sentence (40,073).

Methods

What is the source of the empirical data? i.e., did the authors reach out to individual countries collecting DHS data or is there a repository with publicly available data you utilised? Briefly describe what the frequency and duration of data collection from the various DHS are – i.e., shortest/longest datasets? Is this frequency/duration comparable (consistent) across the 29 countries included, or are they different?

How complete is the empirical data from the countries (please discuss missingness)? Was any imputation necessary?

A sheltered/secure birthing facility (versus not) may be protective against extreme temperature exposures for new-borns. Was information available on the place of birth? If yes, please separate out the place of birth and conduct a sensitivity analysis for the very early neonates.

Analysis

In line 160, it would help if you can provide one sentence to describe how you present the results, i.e., first presenting effect estimates from individual countries, and then meta-analysis of pooled countries.

Figure 2 caption: please say the pooled analysis across '29 countries'

Discussion

Line 137: However, for the overall neonatal period, the risk of mortality was higher at lower temperatures (Fig. 1a and b), while for the very early neonatal period, the risk increased more steeply at higher temperatures (Fig. 1c and d).

Can you describe why you observed this across the 29 regions?

Line 180 and 137: present the absolute numbers of deaths caused by heat and deaths saved from cold. To provide context in the discussion, can you elaborate on how these values compare against the overall size of the neonatal population in these countries?

Line 198: Overall, the impacts of climate change on temperature-related neonatal mortality were largest either in countries with relatively high baseline neonatal mortality rates (e.g., Sierra Leone, Ethiopia, Liberia, 7 Mali) or those that experienced large temperature increases due to climate change (e.g., the Philippines, Uganda, and Rwanda).

Are there any other countries that fit into these two categories? Please discuss what relevance this has to the findings.

Line 248: change to 'prematurely'

Line 257: Climate change impacts on neonatal health have been uneven across regions, with the largest losses from increased heat but also gains from decreased cold observed in countries in sub-Saharan Africa.

Can you discuss how this compares to other regions studied, and why these observations are likely manifesting?

Line 259 to 262: While the decline in the cold-related burden may aid efforts to reduce neonatal deaths, with ongoing climate change, these gains are likely to be outstripped by progressively increasing heat-related mortality in the future.

Can you add a reference here please?

Line 266-269: There is also the seminal study of Gasparrini et al, where authors look at exposure to heat and cold and show (as you do) that exposure to moderate heat and cold have a greater burden than extreme temperatures (in the general population, not neonates). It would be good to discuss your results in the context of this work (i.e., your findings from lines 167 - 170):

[https://www.thelancet.com/journals/lancet/article/PIIS0140-6736\(14\)62114-0/fulltext](https://www.thelancet.com/journals/lancet/article/PIIS0140-6736(14)62114-0/fulltext)

Line 305: current policies in place, climate change is projected to increase xxx to 2.8°C by the end of the...

please add the missing words here indicated as xxx: is it 'ambient temperature' or 'surface temperature'?

Extended Fig 2 and 3: can you please order the countries by continent groupings, rather than alphabetically?

REVIEWER COMMENTS

Reviewer #1 (Remarks to the Author):

This study uses time-stratified case-crossover analysis to quantify the non-linear association between daily ambient temperatures and neonatal mortality and then estimates the attributable fraction of neonatal mortality attributed to non-optimal temperature. In addition, the authors also calculated the excess number of neonatal deaths attributable to climate change. These findings could provide valuable evidence for the impact of climate change on neonatal mortality. However, there are still some concerns that should be addressed by the authors.

Major comments

1. Line 295. It is hard to say that the result derived from this study is global evidence since samples are collected from limited countries mainly in Africa. More importantly, almost half of the deaths were located in India, which weakens the representatives of the main findings.

Response: Thank you for this comment. Indeed, our samples are mainly based on populations in Africa and South Asia. We have now revised the text, so it does not provide misleading information regarding global representation:

This study provides the first evidence of the contribution of historical climate change to temperature-related neonatal deaths. *[Line 347-348]*

The reviewer is correct that the estimation of the exposure-response function (ERF) is likely dominated by India due to its large sample size. We acknowledged that this is an important limitation of the dataset that we are using. Reducing uncertainty in the representativeness of the ERF would require larger sample sizes for all countries, but other neonatal mortality data for these countries are largely missing due to their underdeveloped monitoring systems. We have now specified this limitation in the Discussion section:

Furthermore, generalizability of our findings is weakened by the disproportionate share of India in our sample. Routinely collected data on neonatal mortality from vital registrations or other reliable data sources with geographic coordinates would be needed to model location-specific exposure-response functions more flexibly and provide more generalizable estimates. *[Line 338-341]*

2. Line 336. What is the distribution of the time gap between the occurrence of neonatal deaths and the interview of the mothers? A maximum of 15 years is too long to recall the accurate time of neonatal deaths, which may cause substantial recall bias.

Response: Thank you for pointing this out. We conducted a sensitivity analysis to test for the potential effect of misreporting or inaccurate reporting of deaths on our results due to problems with recall. Restricting the analysis to neonatal deaths reported within 5, 10 and 15 years prior to the interview yielded a similar shape of the ERF and an identical minimum mortality temperature (MMT). However, Confidence Intervals (CIs) for estimates based on the shorter recalls were wider due to the smaller sample sizes. Nevertheless, our ERFs fall within the CIs of the ERFs with shorter recall periods. Please see reply to Comment 6 below for more details on this sensitivity analysis.

In addition, we have now explicitly acknowledged this limitation in the Discussion section and clarified that if there is any misreporting of deaths (including inaccurate date due to too long recall periods)

this is likely to be non-differential and not dependent on daily temperature. Thus, the long recall period potentially adds more noise to the data but it is not likely to bias our final results.

DHS data are based on self-reported birth history, and we included information on neonatal deaths within 15 years of the interview date. Furthermore, DHS data may not fully capture neonatal deaths due to social stigma in reporting, omissions, and mis-recording by the interviewer or misclassification between very early neonatal deaths and stillbirths⁴⁰. However, since case-crossover is a self-matched design, meaning each case is compared to itself, and not others, and misreporting of deaths (including inaccurate date due to too long recall periods) is likely to be non-differential and not dependent on daily temperature, omission and misreporting of neonatal deaths is unlikely to have affected our results. [Line 318-326]

3. Line 361. Indeed, the temperature percentiles may to some extent account for the population adaptation to their local climate. However, this method may lose considerable statistical power, since plenty of information on the temperature was omitted. For example, one of the main impacts of climate change is the substantial increase in extreme heat events, especially in warm regions. However, when using temperature percentiles, the difference between extreme temperatures and moderate temperatures may be narrowed.

Response: Thank you for pointing out this important issue. We agree that this is a limitation of the model that we are using, but unfortunately, we could not use absolute temperatures and apply the standard BLUP method in order to get more precise effects due to our sample size. Please see reply to comment 5 below for more details. We have now acknowledged this limitation in the Discussion:

The use of temperature percentiles might have also led to underestimation of the impact of extreme heat events as the difference between extreme temperatures and moderate temperatures may be narrowed with this method. [Line 331-334]

4. Line 378. More detailed information about the model specification should be presented. For example, Are any confounding factors, such as air pollution and humidity, adjusted for?

Response: We have now included more information on the model specifications, particularly concerning adjustment for potential confounding by air pollution and humidity.

We did not include air pollution in the model because daily data are unavailable. However, the existing literature indicates modest^{20,61,62} or no^{63,64,65} confounding effect of air pollution on the temperature-mortality or -preterm birth associations. We also did not control for relative humidity in the analyses due to minimal observed confounding effects in previous studies^{66,67}. [Line 436-440]

As already clarified in the Discussion, we did not control for any time-invariant characteristics since the case-crossover analyses are case-only, within-person comparisons and therefore the design eliminates confounding by stable individual characteristics. Also, in the time-stratified case-crossover analyses season, month, and day of the week are controlled for by design.

5. The authors combined data from all countries into one model, which implied a fixed effect of temperature on neonatal deaths for all countries. This assumption may be not appropriate, especially when considering that almost half of the samples were from one country (namely India). Plenty of previous epidemiology studies confirmed that the association between temperature and mortality varied by region due to different climate conditions, socioeconomic factors, and so on. Thus, in line with previous studies, I would suggest a two-stage analysis strategy, namely modeling

the impact of temperature for each country and then pooling the effects of all countries by random effect meta method. Since many countries have limited sample sizes, the BLUP method could be applied to re-estimate the country-specific effect by borrowing information from other countries.

Response: Thank you very much for the suggestion. We agree that a two-stage analysis strategy would allow us to more flexibly model the temperature-mortality association. We actually started with this approach as it is the standard method in multi-country analysis on temperature and mortality. However, this method was not applicable in this particular setting – when we ran the meta-analyses, data from countries with a small number of observations were discarded, which led to very large confidence intervals (see Figure 1 below). Therefore, we had to rely on an alternative model, where all observations are pooled together and differences in climate conditions are accounted for by using temperature percentiles instead of absolute temperatures. It is true that with this approach temperature effects on neonatal mortality are fixed and we were not able to model country-specific differences in risk. However, the estimates of average effects are still valid, and we believe they provide a valuable contribution to the literature, given the current lack of evidence and the pressing need to understand the impacts of climate change on populations in LMICs. Furthermore, the two-stage analysis provides similar point estimates to the one-stage analysis (see figure below) despite the wider CIs, which is reassuring. We have already acknowledged in the limitations of our analyses that we cannot obtain country-specific differences in risk.

Figure 1: First-stage country-specific and pooled (95% CI as grey area) summaries from the model on ambient temperature and neonatal mortality, using temperature reanalysis dataset gswp3-w5e5.

6. Sensitive analyses are needed to verify the robustness of the main analyses, especially the exposure-response association between temperature and death.

Response: Thank you for the suggestion. We conducted a number of sensitivity analyses.

First, to test for potential effect of misreporting or inaccurate reporting of deaths on our results due to problems with recall, we re-estimated the association after restricting neonatal deaths to those reported by the mother within the last 5 and 10 years prior to the interview date in addition to the last 15 years as in the main analysis. This resulted in considerably different sample sizes: 11,943 for the 5 years recall period, and 26,272 for the 10 years recall period (compared to 40,073 for the 15 years recall period). As shown in Figure 2 below, analyses by recall period yielded a similar shape and

an identical minimum mortality temperature (MMT). However, CIs for estimates based on the shorter recall are wider due to the smaller sample sizes. Nevertheless, our ERFs fall within the CIs of the ERFs with shorter recall periods.

Figure 2. Sensitivity analysis of overall cumulative temperature–neonatal mortality association based on recall period of the mother, using temperature reanalysis dataset gswp3-w5e5.

Second, we also assessed model sensitivity to different knot placements for the temperature–neonatal mortality association. We examined the following: one internal knot placed at the 10th, 20th, 30th, 50th, 70th, 90th percentiles; two internal knots placed at the 25th and 50th; 50th and 75th; 50th and 90th percentiles and three internal knots placed at the 10th, 50th and 75th percentiles (See Figure 3 below).

The models with one knot yielded similar curves but with slightly different points of minimum mortality, while those with two or three knots (except for the one with knots at the 25th and 50th percentiles) showed a decreasing trend at the higher temperature percentiles. The final model was selected based on the Akaike information criterion (Please see Supplementary Material Table 6).

Figure 3. Sensitivity analysis of the overall cumulative temperature–neonatal mortality association based on knot number and placement, using temperature reanalysis dataset gswp3-w5e5.

7. Line 386. Why authors choose the lag period of up to two days as the maximum lag period of temperature? Previous studies indicated that the impact of cold on mortality may persist for up to weeks. Authors can provide more information about the lagged effect of temperature, especially the difference between two kinds of death since their underlying mechanisms may be different. In addition, it would provide valuable information for the prevention of neonatal death.

Response: Thank you for raising this important methodological question. As we use a time-stratified case-crossover analysis, we are limited in the number of lags that we could investigate. Only short lags (0-7 days) are allowed for this type of analysis as longer lags would overlap with the selected controls, which are separated by the outcome by only 7 days. We conducted additional analysis to assess specifically the lagged effects of temperature within these 7 days window of exposure. In particular, we estimated and plotted the effects for specific temperature percentiles and lags.

Figure 4 below shows the RRs for very early neonatal mortality by temperature percentile at specific lags (0,2,4,7) and by lag at specific temperature percentiles (2.5th, 25.5th, 75.5th, 97.5th), corresponding for the selected cut-offs for moderate and extreme heat. The plot confirms more delayed effects for the extremely cold and hot temperatures (1st and 4th graph on the right) compared to the moderately cold and hot ones, with significant risk lasting up to 2 days. Effect estimates for longer lag periods were not significant and therefore, we only considered the 2 days lag period in our final analysis.

Figure 4: Plot of RR for very early neonatal mortality by temperature at specific lags (left) and RR by lag at 2.5th, 25.5th, 75.5th and 97.5th percentiles of temperature distribution (right), using temperature reanalysis dataset gswp3-w5e5.

Figure 5 below shows the RRs for neonatal mortality by temperature percentile at specific lags (0,2,4,7) and by lag at specific temperature percentiles (2.5th, 25.5th, 75.5th, 97.5th). The plots show more delayed effects only for the extremely cold temperatures (1st graph on the right in Figure 5) compared to the moderately cold and moderately and extremely hot temperatures, with significant risk lasting up to 2 days. Effect estimates for longer lag periods were not significant and therefore, we only considered the 2 days lag period in our final analysis. As the risk estimates fall short of significance just before the 3 days lag period we conducted a sensitivity analysis by estimating cumulative RR with 2- and 3-days lag. As shown on Figure 6 below (left and right panel) the two exposure-response curves were largely similar.

Figure 5: Plot of RR for neonatal mortality by temperature at specific lags (left) and RR by lag at 2.5th, 25.5th, 75.5th and 97.5th percentiles of temperature distribution (right), using temperature reanalysis dataset gswp3-w5e5.

Figure 6: Overall cumulative temperature–neonatal mortality associations for A) Lag 0-2 and B) Lag 0-3, using the temperature reanalysis dataset gswp3-w5e5.

8. Line 406. Is it the “cut-offs corresponding to the 97.5th, 75.5th, minimum mortality temperature, 25.5th, and 2.5th temperature percentiles”?

Response: Yes, this is correct. Thank you! We have now corrected it in the manuscript.

9. The authors calculate the excess fraction of neonatal deaths attributable to climate change by simply subtracting the hot- and cold-related burdens between the factual and counterfactual scenarios. This calculation completely ignores human adaptation to temperature changes. As the authors indicated in the manuscript, people can adapt to local climate conditions, so they may also adapt to local climate changes. Thus, the excess fraction of neonatal deaths attributable to climate change may be overestimated (for heat).

Response: Thank you for this comment. First, it is important to distinguish between intrinsic and extrinsic adaptation. Intrinsic adaptation occurs as a result of a *physiological acclimatisation* response within the population to changes in temperature and normally does not appear in a short period of time. Extrinsic adaptation, on the other hand, encompasses *planned interventions* and *behavioural change* in response to changing conditions as well as *spontaneous adaptation*. The latter occurs through non-climate driven factors such as socioeconomic development, improved health care services or infrastructural development.

Since we consider an 18-year period in our analysis we believe that this time frame is too short for a physiological acclimatisation to higher temperatures to occur. It is true that we do not consider variation in the ERF in the study period that might have occurred due to planned interventions or behavioural adaptation, but our model will be underpowered to pick up such variation. Furthermore, adaptational responses such as heat-health early warning systems and air conditioning have relatively low uptake in the studied countries^{1,2,3} and we expect a small impact on our estimates.

Differences in susceptibility to heat and cold between the factual and counterfactual scenario are likely to be dominated by spontaneous adaptation. However, spontaneous adaptation normally occurs simultaneously but independently from the changing climate. Therefore, while it is true that the ERF might change over time, this change is not likely to be so much a result of warming but of decreasing vulnerability, which could have happened even in the absence of climate change.

One could argue that spontaneous adaptation is indirectly linked to warming through the close link between economic growth and improvements in living standards with increases in GHGs observed historically. Hence, susceptibility to heat and cold might have been higher in a counterfactual scenario where general standards of living and healthcare may have remained stagnant. However, this is difficult to predict and quantify given the complex multi-causal processes involved and the lack of empirical data. The demographic profile of the population in such a scenario may have also been different. Given the complexity of modelling such a hypothetical scenario, our analysis has focused on isolating impacts of climate change-induced increases in temperature only, while keeping socio-economic development constant in both the factual and counterfactual scenario.

We included one sentence in the Discussion, acknowledging that changes in the exposure-response function over time due to adaptation is not considered:

We also did not consider changes in the exposure-response function over time due to adaptation, but such changes are likely to be dominated by non-climate driven factors such as socioeconomic development, improved health care services or infrastructural development, which occur simultaneously but independently from the changing climate. [Line 334-338]

1. Mastrucci, A., Byers, E., Pachauri, S. & Rao, N. D. Improving the SDG energy poverty targets: Residential cooling needs in the Global South. *Energy Build.* **186**, 405–415 (2019).
2. Li, T., Chen, C. & Cai, W. The global need for smart heat–health warning systems. *Lancet* **400**, 1511–1512 (2022).
3. Kotharkar, R. & Ghosh, A. Progress in extreme heat management and warning systems: A systematic review of heat-health action plans (1995-2020). *Sustain. Cities Soc.* **76**, 103487 (2022).

Reviewer #2 (Remarks to the Author):

General comment:

This is very well written paper with appropriate methodology applied that links the important topic

of climate change and neonatal mortality burden in low- and middle-income countries.

Response: Thank you for this positive feedback.

Comments

10. It would greatly benefit the readers if the authors explained in detail how 'non-optimal' or minimum mortality temperature percentile was calculated.

Response: Thank you for this comment. We have now included a more detailed description of how the minimum mortality and the non-optimal temperature percentiles were defined in our analysis:

To determine the temperature of minimum mortality we generated predictions of the RR on a matrix of values of the original predictor (temperature percentiles) and lags using the `dlm` package. We then selected as minimum mortality or optimal temperature the temperature percentile value where the predicted RR was minimised. Temperature percentiles below and above that value were considered non-optimal. [Line 452-456]

11. I wonder if the authors could access data from the health and demographic surveillance sites in LMIC implementing verbal autopsy to corroborate the findings?

Response: Thank you for this interesting suggestion. Unfortunately, HDSS datasets are not freely available but we are currently applying for funds to work with some HDSS sites and it would definitely be interesting to try to corroborate our findings with these data. DHS also conducts verbal autopsies for certain countries. However, currently verbal autopsy data from DHS are available for only 2 of the 37 DHS surveys included in our analysis, representing only 2 of the 29 included countries (namely Bangladesh and Nepal). Since the verbal autopsies are conducted following the DHS survey, the reported dates of birth and death are the same and hence we could not use them to validate reported date of birth and death of each child. Unfortunately, the limited number of countries with available verbal autopsy data also does not allow us to analyse the association between ambient temperature and cause-specific mortality in order to better understand the mechanisms by which ambient temperature triggers neonatal mortality.

12. Why was the data pooled yet country-specific analysis could be done?

Response: As explained in response to comment 5 by Reviewer 1 (Please see above), we had to pool the data for all countries as the limited number of observations in many countries prevented us from doing a two-stage analysis (i.e. modelling country-specific effects and then combining them in a meta regression) as is the standard in temperature-mortality studies across different locations. Since we estimated the temperature-mortality relation on a relative scale, where each temperature percentile corresponds to a different absolute temperature for each PSU, and since temperature distributions between countries differ, it follows that the relative risks across percentiles will be the same, but the minimum mortality temperature and the curves on the original degree scale will differ. Considering these differences, we thus still estimate impacts at country level. Apart from the exposure-response function, variations in impacts across countries also stem from factors such as baseline neonatal mortality rates and level of warming, which we have considered when estimating impacts at country level.

13. Would the results differ if you did country-specific analysis for the exposure response curves and then combined the results later in a meta regression?

Response: Thank you for this suggestion. It is true that applying the two-stage analysis would have allowed us to derive more precise country-specific estimates. By defining the temperature-mortality

association on a relative scale we consider differences in temperature distribution and minimum mortality temperature across locations, but the shape of the exposure-response functions (ERFs) is assumed to be the same. Furthermore, as pointed out by Reviewer 1 in Comment 3, the use of temperature percentiles might have also led to underestimation of the impact of extreme heat events as the difference between extreme temperatures and moderate temperatures may be narrowed with this method. However, it is reassuring that when applying the two-stage analysis we get similar point estimates as for the one-stage analysis, but with wider CIs due to the discarded observations. Please see our response to Comment 5 by Reviewer 1 above, where we provide the ERF from the two-stage analysis and explain in more details the limitations of our approach. We acknowledge these limitations of our analysis in the Discussion, see lines 331-335.

14. Would it matter to account for country specific factors that influence neonatal outcomes in a multilevel kind of analysis?

Response: Thank you for the suggestion. It would indeed be interesting to test for country-specific factors that may influence the temperature effects on neonatal mortality by adding interactions with country-level predictors. However, there is limited power to address this issue in this analysis due to the small sample sizes per country. It will definitely be interesting to investigate this in the future when more data become available.

15. You mention, community specific percentiles, what are the communities in this respect?

Response: We are sorry for the confusion. Communities in this respect refers to the DHS Primary Sampling Units (PSUs) or clusters. In DHS, PSUs represent villages in rural areas and census enumeration blocks in urban areas. We have now changed the text to refer to “location-specific” temperature percentiles

We standardized absolute temperatures in location-specific temperature percentiles to account for population adaptation to their predominant climate. *[Line 87-89]*

In addition, we have described in more details in the methods section what a PSU/cluster is and how the temperature percentiles were defined:

Since we combined data from populations spanning different climate zones, we converted the absolute temperatures into PSU-specific temperature percentiles, following previously described methods. This approach allowed us to account for population adaptation to their predominant climate. We linked the individual cases of neonatal death with the daily mean ambient temperature data using the geographic coordinates of each PSU. *[Line 414-418]*

16. What would be the best mitigation strategies to reduce the burden?

Response: Thank you for this comment. We agree it would be good to add this and have now included one paragraph in the Discussion on public health interventions to reduce the burden of temperature-related neonatal deaths:

Our findings highlight the need for public health interventions to protect newborns in LMICs from low and high ambient temperatures. Prevention of postnatal thermal loss in low-resource settings is a recognized challenge, which has been linked to inadequate thermal control practices (i.e., insufficient heating of the birthplace, placing of the uncovered newborn on the ground or other cold surfaces, delayed wrapping, early bathing.), insufficient knowledge of hypothermia diagnosis, lack of robust and affordable incubators and other infant medical equipment and unreliable power supply⁴³. On the other hand, the risk of heat stress in neonates can be exacerbated by poor postnatal practices (placing newborns in direct sunlight without shading, over-bundling in clothes and blankets), insufficiently frequent breastfeeding in hot weather leading to dehydration, failure of essential infant equipment and

power outages in medical facilities during heatwaves⁴⁴. Effective behavioral and community interventions to reduce mortality risks from neonatal hypothermia and hyperthermia include educational programs among mothers and other caregivers and training of health providers on practical measures that ensure thermal stability (e.g., immediate drying, skin-to-skin contact, delayed bathing)^{44,45,46}. The provision of essential infant devices such as incubators and infant warmers tailored to the context in LMICs are important for preventing cold stress⁴³, but can be prohibitively expensive. Initiation of kangaroo mother care immediately after birth is an alternative low-cost warming method, which has proven effective in reducing neonatal hypothermia, severe infections, and mortality^{47,48}. Apart from good postnatal thermal practices, improving the designs of homes and health facilities, increasing urban green spaces, and providing support for home cooling are some strategies that can help reduce heat stress for pregnant women and newborns⁴⁴. It is important to note that these actions should be part of a broader effort to reduce neonatal deaths, given that temperature-related deaths constitute less than 5% of neonatal mortality according to our findings. [Line 293-317]

17. I wonder how this study based on the design accounts for other competing risk factors in these communities such as high malaria burden, health systems etc

Response: Thank you for this important question. The case-crossover study design is used to investigate associations between short-term environmental exposures and health outcomes, where the effect on risk is immediate. In this study design each case serves as his/her own control, whereby exposure of the same individual on a case day is assessed against referent exposures on days before or after the case day. Therefore, factors that are constant or stable within a short period of time are automatically adjusted for (e.g. access to healthcare, behavioural risk factors). In addition, the selection of control days from the same day of the week, month, and year as the case day also ensures that potential time-varying confounders such as day of the week, seasonality, and long-term trends are accounted for. Since we compare exposure to temperature on the day of the event with nearby days (from one to three weeks before or after the case day), slower onset or longer lasting risk factors such as elevated malaria prevalence and disruptions in health care systems are not likely to affect our results. Studies on temperature and malaria incidence, for instance, tend to look at monthly temperatures and include lags of several months. Extreme heat might potentially have some immediate effects on healthcare systems by affecting health seeking behaviour (e.g. less likely to travel long distance to a health facility), overwhelming health systems (e.g. higher emergency department visits during heatwaves) or affecting postnatal care in hospitals through power outages. However, these factors would be on the causal pathway between ambient temperatures and neonatal mortality (i.e. mediating factors) and should, therefore, not be considered as confounders.

18. So many countries are not represented, could this tilt the outcome if there were more countries with DHS data?

Response: Thank you for this comment. We acknowledge that our results might not be representative for all LMICs since we could not include many countries due to data availability. Therefore, we used the estimated exposure-risk functions to assess the temperature-related burden of climate change on neonatal mortality only for those countries that we had data on and, unlike other studies, did not extrapolate the risk functions to different populations. As DHS surveys with exact date of birth and death of children and geographic coordinates become available for more countries, it will be important to evaluate their ERFs and see how they compare with our results.

19. Can the attribution be extended to the future e.g 2030s ,2050s based on projections data?

Response: Thank you for this suggestion. We are planning another study where we use the estimated ERFs in order to project impacts into the future under different climate change scenarios. We decided to do this in a separate study, since we would like to do a more comprehensive analysis for the future, considering different socio-economic scenarios and potentially the feedback effects of ambient temperature on baseline mortality in the population.

Reviewer #3 (Remarks to the Author):

Temperature-related neonatal deaths attributable to climate change

This is a very interesting study and the findings are meaningful and useful. The results fill in a gap in current knowledge about temperature effects on neonates, where empirical data is being used to calculate the climate attributable burden of neonatal mortality. Presenting the % change in deaths from heat and cold and associated number of deaths provides policy relevant findings. It's great to see countries in Africa and Asia are predominantly featured, where evidence is lacking – and the country specific effect estimates are useful for policymaking/guiding uptake of local adaptation measures. I thank the authors for publishing this work.

Response: Thank you for this positive and encouraging feedback.

20. My concern is that there is an uneven mix of countries making up the 'meta-estimate' grouping when I stratified the 29 countries across continents (Africa, Asia, Caribbean, Europe), i.e., Europe has one country, while Africa has 18. How was this taken into account in the meta-analysis, as the bias in sampling countries may place limitations on comparing effects across continents for the meta-estimates? Can you please explain your approach and how you dealt with the heterogeneity in terms of providing representative meta-estimates.

Response: Thank you for this comment. We were actually not able to conduct a two-stage analysis by modelling country-specific effects and then pooling them in a meta regression as is the standard approach in multi-country temperature-mortality studies. As explained in response to Comment 5 from Reviewer 1 above, this was not possible due to the small number of observations for some countries (please see above for more details). Therefore, we had to use an alternative approach where we pool all the observations together, standardise absolute temperatures for each primary sampling unit (PSU) into PSU-specific temperature percentiles and estimate the temperature-mortality relation on a relative scale. This approach allowed us to still account to some extent for population adaptation to their local climates by considering differences in temperature distributions and minimum mortality temperatures across populations. We acknowledge the limitations of this method in the Discussion section.

In addition, please note that our estimates are not supposed to be representative for each continent. The continent grouping on Figure 3 and 4 in the manuscript was done only for the purpose of ordering the country-level results and making them easier to locate. To avoid any misunderstanding, we now refer to differences between individual countries rather than regions/continents:

Excess heat due to climate change has affected neonatal mortality rates across all study countries, with the largest increases in heat-related rates (>30 per 100,000) observed in Sierra Leone, Ethiopia, Liberia and Haiti (Fig. 3d). [Line 201-203]

The largest increases in heat-related very early neonatal mortality rates induced by climate change (>32 per 100,000) were observed in Liberia, Sierra Leone, Ethiopia and Angola (Fig. 4d). Conversely, cold-

related very early neonatal mortality rates averted due to climate change exceeded 37 per 100,000 live births in Liberia, Rwanda and Uganda, (Fig. 4c). [Line 240-243]

21. The study also misses large LMIC demographics, for example, in Latin America. Can you please justify this? This is a limitation and should be talked about in the discussion.

Response: Thank you for pointing this out. As illustrated on Supplementary Figure 2, we selected DHS surveys for inclusion in our analysis based on the following criteria (i) date of birth and death data were collected; (ii) geographical coordinates of the PSUs were provided; (iii) temperature data were available for least 2 of the 3 ISIMIP datasets. We have now explicitly specified the inclusion criteria for DHS surveys in the Methods section:

We restricted our analysis to surveys with available (i) global positioning system (GPS) data, i.e., latitude and longitude, of primary sampling units (PSUs); (ii) detailed information on the date of birth and death (day, month, and year) and (iii) temperature data for least 2 of the 3 ISIMIP datasets. [Line 373-376]

The reason for excluding countries in Latin America, with the exception of Haiti, was that they had only older DHS survey rounds available, where the exact day of birth or death information were not collected. We have now specified this limitation in the Discussion section:

Also, certain regions, such as Latin America, are largely underrepresented in our analysis due to the lack of recent DHS survey rounds with detailed day of birth and death information there [Line 339-341].

22. Furthermore, a country like Jordan is strictly speaking in the middle-east, however, this is not a grouping presented in this study (I understand why you may have grouped it into Asia, where you have more countries). How were the groupings created? Rather than arbitrarily assigning groupings by continent, grouping by a climate classification such as the Koppen Geiger classification can enable effect estimates from countries with more homogeneous climates to be compared and meta-analysed. Can you please discuss your use of grouping by continent, and why this was selected (and what the pros and cons are)?

Response: Thank you for this comment. As explained in response to Comment 20 above, the country groupings on Figure 3 and 4 were done only for the purpose of ordering country-level results and making them easier to locate. We pooled the data for all countries together to estimate the exposure-response function (ERF) on a relative scale and then calculated the impact for each country based on it. We did initially try to group the DHS clusters based on their climate zone using the Koppen Geiger classification and to pool results together in a meta-analysis. However, for many of the climate zones we could not estimate the effect due to an insufficient number of observations. Therefore, we decided to pool observations of all countries together and estimate the ERF on a relative scale. For ordering our results it would be difficult to group the countries by climate zone as many countries encompass several climate zones.

23. I can see in line 321 you give the rationale for including 29 out of 90 countries collecting DHS data as “We restricted our analysis to surveys for which global positioning system (GPS) data, i.e., latitude and longitude, of primary sampling units (PSUs) were available”. Was this the only rationale, or was it the availability/completeness of the data?

Response: We selected DHS surveys for inclusion in our analysis based on the following criteria (i) date of birth and death data were collected; (ii) geographical coordinates of the PSUs were provided; (iii) temperature data were available for least 2 of the 3 ISIMIP datasets. We have now explicitly specified the inclusion criteria for DHS surveys in the Methods section. Please see lines 373-376.

24. A general comment on style: I find active voice to be more effective and engaging for the reader – it would be nice if the authors can consider using this in future publications.

For example line 225: On the other hand, climate change was estimated to have reduced cold-related very early neonatal mortality by 35% (range: 10-69%), amounting to 141,322 fewer neonatal deaths (95% UI: 2,377-339,337) (see Fig. 4e and Supplementary Table 3 for country-specific estimates).

Could be rephrased as: We found that climate change reduced cold-related very early neonatal mortality by 35% (range: 10-69%), amounting to 141,322 fewer neonatal deaths (95% UI: 2,377-339,337) (see Fig. 4e and Supplementary Table 3 for country-specific estimates).

Response: Suggestion accepted. We have revised this sentence and various others in the manuscript and will use active voice more often in future publications.

Please find my specific comments attached below:

Title and abstract

25. Title: In the title please include: (i) sample of neonatal deaths analysed (40,073); (ii) a statement such as ‘an analysis of 29 low- and middle-income countries’ to indicate the scope.

Response: Thank you for the suggestion. We have now changed the title to “Temperature-related neonatal deaths attributable to climate change in 29 low- and middle-income countries”. However, Nature Communications requires that “the title should be 15 words or fewer”. Therefore, we do not have sufficient word count to include also the sample size as suggested, but we included it instead in the abstract.

26. Line 11: 29 low- and middle-income countries

Please add the following text before 29 so the sentence reads: Demographic and Health Surveys (DHS) data from 29 low- and middle-income countries.

Response: Suggestion accepted

27. Line 15 between “30%. [insert here.] Countries” please insert effect estimates on the range of mortality burden attributable to temperature between individual countries you included (i.e., the individual country that had the highest and lowest burden)?

Response: Suggestion accepted. We have now specified in the abstract the country range for the heat-related and cold-related neonatal mortality burden attributable to climate change. Please note that the number for heat-related mortality changed since before we referred to the percentage increase in heat-related mortality from the counterfactual scenario. As suggested by the reviewer the text now refers to the fraction in the factual scenario attributable to climate change.

Climate change was responsible for 32% (range: 19-79%) of heat-related neonatal deaths, while reducing the respective cold-related burden by 30% (range: 10-63%). [Line 15-17]

28. Line 15: Countries in sub-Saharan Africa have experienced the most pronounced climate-induced losses from increased heat and gains from decreased cold

In the results component of the abstract there is no indication of effect estimates (% deaths) presented for region groupings other than Africa (other than ‘all countries’). This could be interpreted as the authors only reporting positive or significant results in the abstract. For Africa, please describe the % of deaths attributable for heat and cold. Please also describe effect estimates (% deaths) from meta-analysis of other regional groupings. Furthermore, please clearly describe how the 29 countries were grouped for meta-analysis (was it by continent, by Koppen Geiger

classification?). A short sentence on this would add clarity (I see from further reading that it's by continent).

Response: Thank you for this suggestion. As clarified in response to Comment 20 above, we did not conduct a meta-analysis by continent or any other grouping. For deriving the exposure-response function (ERF), we pooled data from all countries together, but we then applied the ERF to the temperature and neonatal mortality data in each country separately to estimate country-specific impacts.

To avoid confusion, instead of referring to regions throughout the manuscript, we now do a comparison of individual countries. In the abstract, we now specify that climate change impacts were observed in all study countries, but largest impacts were observed in countries in Africa. Due to the word limit for the abstract (150 words) we could not include more detailed information. However, we have now included also the country ranges of observed heat and cold impacts as suggested.

29. Please conduct the meta estimates of the associations. Please include meta estimates in figures 3 and 4 with the extended figures 2, 3 4 and 5.

Response: Thank you for this suggestion. We were actually not able to conduct a two-stage analysis by modelling country-specific effects and then pooling them in a meta regression as is the standard approach in multi-country temperature-mortality studies. As explained in response to Comment 20 and to Comment 5 by Reviewer 1 above, this was not possible due to the small number of observations for some countries. Therefore, we had to use an alternative approach where we pool all the observations together, standardise absolute temperatures for each PSU into PSU-specific temperature percentiles and estimate the temperature-mortality relation on a relative scale. Please see response to Comment 5 above for more details.

30. Line 19: the most vulnerable populations.

Considering your population of interest are neonates, please be specific and use 'neonates' here, rather than speaking generally about vulnerable populations.

Response: We have now used newborns instead of "vulnerable populations".

Introduction

31. Lines 36 to 42: please use consistent terminology - neonates or infants (not interchanging use both):

Response: Suggestion accepted.

32. Line 71: increase awareness of what?

Response: Thank you, we have now specified:

Impact attribution studies can increase awareness of the already occurring impacts of climate change.. [Line 72-74]

33. Line 72: inform societal responsibility for the cost of damages and losses. I would change this sentence to "inform society about costs and damages due to climate change"

Response: Thank you for the suggestion, we have revised the sentence accordingly.

34. Line 79-81: we combined the largest internationally comparable dataset on neonatal deaths in LMICs with three recent temperature reanalysis datasets and corresponding counterfactual datasets

in a two-stage analysis.

Please state what the empirical ‘largest internationally comparable dataset’ is.

Response: We have now included a reference to a publication to support this statement. The publication refers to other existing survey platforms used for estimating child mortality and highlights DHS as the “the main source of information on child mortality in most low- and middle-income countries over the past three and a half decades”.

Akuze, J. et al. Four decades of measuring stillbirths and neonatal deaths in Demographic and Health Surveys: historical review. Popul. Health Metr. 19, 1–14 (2021).

Further empirical justification is provided in the Methods section:

Since its launch in 1984, the DHS program has conducted surveys in more than 90 low- and middle-income countries (LMICs), systematically collecting data on neonatal deaths, and is thus the largest publicly available source for such data³² [371-373]

35. Line 77: to the burden of temperature-related neonatal deaths in 29 LMICs over the period Please state the total sample size of deaths analysed in the sentence (40,073).

Response: Suggestion accepted.

Methods

36. What is the source of the empirical data? i.e., did the authors reach out to individual countries collecting DHS data or is there a repository with publicly available data you utilised? Briefly describe what the frequency and duration of data collection from the various DHS are – i.e., shortest/longest datasets? Is this frequency/duration comparable (consistent) across the 29 countries included, or are they different?

Response: The source of the empirical health data is the data repository of the DHS Program. The data are publicly available for academic research, but prior registration is required. DHS are typically conducted every 5 years, each time with a different cross-sectional sample, but for our analysis we were restricted to DHS from the last survey rounds (DHS-VII and DHS-VIII) as earlier survey rounds did not collect information on exact date of birth and death of children. The average duration of data collection for the included surveys was 6 months. However, the collection period between countries differed, ranging from 3 months in the Philippines to 18 months in India. As specified in the Methods section we included neonatal deaths reported by the mother within 15 years prior to the interview date.

37. How complete is the empirical data from the countries (please discuss missingness)? Was any imputation necessary?

Response: From all the selected DHS surveys we derived 72,756 cases of neonatal deaths. Of these 432 observations (0.6%) had missing geographic coordinates and 98 observations (0.1%) had missing climate data. No imputation was necessary due to the small number of missing observations. We further restricted the neonatal deaths to be included in the analysis to those reported in the last 15 years, which resulted in 40,073 observations in the final analysis. We have now integrated this information in the flow chart in Supplementary Figure 2.

38. A sheltered/secure birthing facility (versus not) may be protective against extreme temperature exposures for new-borns. Was information available on the place of birth? If yes, please separate out the place of birth and conduct a sensitivity analysis for the very early neonates.

Response: Thank you for this suggestion. It may be indeed interesting to check whether the effect of ambient temperature differs between those born in a sheltered/secure birthing facility and not. However, we do not have sufficient statistical power to test for this. Furthermore, even if stratified analysis by place of birth would suggest effect modification this would not change our overall effect estimates.

Analysis

39. In line 160, it would help if you can provide one sentence to describe how you present the results, i.e., first presenting effect estimates from individual countries, and then meta-analysis of pooled countries.

Response: As clarified above we did not conduct a meta-analysis, but applied the same ERF to estimate impacts for individual countries. Country-level impacts are summed up to arrive at overall estimates for all included countries

40. Figure 2 caption: please say the pooled analysis across '29 countries'

Response: Suggestion accepted.

Discussion

41. Line 137: However, for the overall neonatal period, the risk of mortality was higher at lower temperatures (Fig. 1a and b), while for the very early neonatal period, the risk increased more steeply at higher temperatures (Fig. 1c and d).

Can you describe why you observed this across the 29 regions?

Response: In the Discussion section we explain potential reasons for observing these differences between the very early neonates and all neonates. More specifically, we hypothesise that higher temperatures more strongly affect very early neonatal mortality due to the effect of heat on pre-term birth and other birth complications, which are leading causes of early neonatal mortality. The higher susceptibility of later neonates to cold, on the other hand, could be potentially explained by their higher incidence of severe infections, which are generally associated with cold.

This may be explained by the different causal pathways through which ambient temperatures might affect neonatal mortality at different periods. Neonates who die within 24 hours after birth are more likely to be infants born after complications during childbirth and prematurity^{11,12}. Prematurity and other birth complications have been consistently associated with exposure to non-optimal ambient temperatures in utero, particularly heat^{13,14,15}. Neonates who survive the first 24 hours, on the other hand, are more likely to die from causes related to severe infections such as sepsis and pneumonia^{11,12}, which have generally been associated with hypothermia^{35,6}. [Line 260-267]

42. Line 180 and 137: present the absolute numbers of deaths caused by heat and deaths saved from cold. To provide context in the discussion, can you elaborate on how these values compare against the overall size of the neonatal population in these countries?

Response: Suggestion accepted. We have now included a description of how climate-attributable burdens vary across countries as a fraction of countries' total (very early) neonatal mortality:

In terms of fraction of all neonatal deaths, the heat-related burden attributable to climate change ranged from 0.2% in Armenia to 1.1 % in Haiti, while the averted burden from cold ranged from 0.3% in Albania, Nepal and Tajikistan to 4.6% in the Philippines (Extended Data Fig. 4 and Supplementary Table 4). [Line 205-209]

In relation to all very early neonatal deaths in a country, the heat-related impact from climate varied from null in Armenia to 2.9% in Haiti. Concurrently, the cold-related fraction mitigated by climate change ranged from 0.2% in Albania, and Tajikistan to 3.7% in Uganda (Extended Data Fig. 5 and Supplementary Table 5). [Line 243-247]

43. Line 198: Overall, the impacts of climate change on temperature-related neonatal mortality were largest either in countries with relatively high baseline neonatal mortality rates (e.g., Sierra Leone, Ethiopia, Liberia, 7 Mali) or those that experienced large temperature increases due to climate change (e.g., the Philippines, Uganda, and Rwanda).

Are there any other countries that fit into these two categories? Please discuss what relevance this has to the findings.

Thank you for pointing this out. We have rephrased the sentence to reflect that we refer to the combination of both factors – large temperature increases and high baseline mortality rates – and we have added other countries that fall into this category as suggested:

Overall, the impacts of climate change on temperature-related neonatal mortality were largest in countries that had relatively high baseline neonatal mortality rates and at the same time experienced large temperature increases due to climate change (Sierra Leone, Ethiopia, Liberia, Mali, Guinea, Benin, Cameroon, Nigeria, Angola, Timor-Leste, Haiti). [Line 209-213]

The relevance of this observation is that both baseline disease burden and absolute change in temperature are important for determining impacts of climate change. Since we estimate both impacts from heat and cold it is not straightforward to determine what the overall impact of climate change would be for a country. We refrained from reporting net impacts (increases in heat mortality minus decreases in cold mortality) as the goal of public policy should be to decrease overall neonatal deaths as opposed to decreasing cold-related at the expense of increases in heat-related mortality.

44. Line 248: change to ‘prematurely’^

Response: In this sentence we refer to the noun “prematurity” and not the adjective, so we have kept the text as it is.

45. Line 257: Climate change impacts on neonatal health have been uneven across regions, with the largest losses from increased heat but also gains from decreased cold observed in countries in sub-Saharan Africa.

Can you discuss how this compares to other regions studied, and why these observations are likely manifesting?^

Response: Thank you for this comment. Following other reviewers’ comments, we decided not to make a direct comparison between regions as this might be misleading due to the fact that some of them are represented by a single country only. Now we focus instead on comparison between individual countries. We have included a sentence in the Discussion pointing out the differences in baseline neonatal mortality rates and levels of warming as the main underlying reasons for the observed geographical variations in climate impacts:

Climate change impacts have been uneven, with the largest losses from increased heat but also gains from decreased cold observed in countries that had high baseline neonatal mortality rates and experienced the most warming. [Line 273-276]

46. Line 259 to 262: While the decline in the cold-related burden may aid efforts to reduce neonatal deaths, with ongoing climate change, these gains are likely to be outstripped by progressively

increasing heat- related mortality in the future.

Can you add a reference here please?

Response: Thank you for the suggestion, we have now included the following two references, which show that for warmer temperatures, low- and middle-income countries' heat-related mortality is projected to exceed cold-related mortality under high emission scenarios, leading to a net increase in temperature-related deaths.

Gasparrini, A. et al. Projections of temperature-related excess mortality under climate change scenarios. Lancet Planet. Heal. 1, e360–e367 (2017).

Bressler, R. D., Moore, F. C., Rennert, K. & Anthoff, D. Estimates of country level temperature-related mortality damage functions. Sci. Rep. 11, 1–10 (2021).

47. Line 266-269: There is also the seminal study of Gasparrini et al, where authors look at exposure to heat and cold and show (as you do) that exposure to moderate heat and cold have a greater burden than extreme temperatures (in the general population, not neonates). It would be good to discuss your results in the context of this work (i.e., your findings from lines 167 – 170): [https://www.thelancet.com/journals/lancet/article/PIIS0140-6736\(14\)62114-0/fulltext](https://www.thelancet.com/journals/lancet/article/PIIS0140-6736(14)62114-0/fulltext)

Response: Thank you for this suggestion. We have now included a sentence on this in the Discussion, also referring to the publication above:

Similar to other studies³⁵, we find that moderately hot and cold temperature dominate the temperature-related burden, which could be explained by their higher frequency throughout the year. [Line 254-257]

48. Line 305: current policies in place, climate change is projected to increase xxx to 2.8°C by the end of the...

please add the missing words here indicated as xxx: is it 'ambient temperature' or 'surface temperature'?

Response: Thank you for pointing to this. We have now revised the sentence as follows to make clear we refer to ambient temperature increases.

With current policies in place, global average surface temperature is projected to increase by the end of the century to 2.8°C above pre-industrial levels⁴⁴ compared to the current 1.1°C, which is likely to lead to progressively increasing heat-related burdens and diminishing cold-related gains. [Line 356-359]

49. Extended Fig 2 and 3: can you please order the countries by continent groupings, rather than alphabetically?

Response: Suggestion accepted and implemented.

REVIEWER COMMENTS

Reviewer #1 (Remarks to the Author):

The authors have attempted to address my comments. But here, I have some other comments or concerns for the authors.

1. For the methodology section, as case-crossover design is used for each individual case and the odds ratio is computed by comparing the intensity of temperature exposure during case period and during the reference period, I am quite confused about the calculation of attributable burden of neonatal mortality to the temperature exposure. A further explanation and rational support on this calculation is required.
2. The authors have mentioned "up to seven days of lag" that were used in modelling choice. But it seems that only the result of AIC on lag 0-2 days is provided.
3. Considering the huge heterogeneity of risk of temperature across the country, when calculating the total effect across country, it is better to include the country variable as a random-effect variable in the model.
4. In comparison with the temperature percentile, absolute temperature is more familiar to the public and policymakers. Therefore, I will suggest some key results using absolute temperature.

Minor comment:

1. The "relative risk (RR)" reported in the method and result section should be the "odds ratio (OR)".
2. For the period of temperature-related burden estimation, "2000-2019" is more common in previous studies than "2001-2019". In addition, for the cutoffs to define moderate cold and heat, I will suggest using "25th" and "75th" rather than "25.5th" and "75.5th".
3. Line 166, this subtitle should be "Temperature-related neonatal deaths attributable to climate change"
4. Line 380, surveys were collected only between 2014 and 2021, which seems to conflict periods (5 and 10 years prior to the interview) mentioned in Line 461.
5. Lines 458-459, please specify the information on the number and placement of knots for exposure-response function.
6. Correlation test is required for the Extended data Figure 1.

Reviewer #1 (Remarks on code availability):

The authors have not provided the code for reviewing.

Reviewer #2 (Remarks to the Author):

The authors have sufficiently addressed my comments.

Reviewer #3 (Remarks to the Author):

The authors have adequately addressed my comments and suggestions. Thank you for the work on improving the manuscript. I recommend the manuscript for publication.

Reviewer #3 (Remarks on code availability):

There is no code available to review.

REVIEWER COMMENTS

Reviewer #1 (Remarks to the Author):

The authors have attempted to address my comments. But here, I have some other comments or concerns for the authors.

1. For the methodology section, as case-crossover design is used for each individual case and the odds ratio is computed by comparing the intensity of temperature exposure during case period and during the reference period, I am quite confused about the calculation of attributable burden of neonatal mortality to the temperature exposure. A further explanation and rational support on this calculation is required.

Response: We apologise if our explanation was not sufficiently clear. The reviewer is correct that in the case-crossover design temperature on the day when an individual event occurs (case day) is compared to temperature on nearby days (control days) in order to determine whether events are associated with a particular exposure (Maclure, 1991). In our analysis each case was matched with up to four control days, selected within the same month and matched on day of the week, following a time-stratified sampling scheme (Navidi, 1998). Please note that since we apply a logistic regression in a time-stratified case-crossover design the coefficients derived from the regression analysis represent (log) RR, not OR.

After calculating the overall cumulative temperature-mortality association we utilise a backward approach to calculate the attributable fraction (AF) of neonatal deaths associated with non-optimum temperature (Gasparrini & Leone, 2014). This method allows to incorporate the added temporal dimension in exposure-response relationships with complex temporal patterns and has been extensively applied in past research using the DLNM framework (Fu et al., 2018; He et al., 2023).

More specifically, for each case day of the sample the attributable fraction $AF_{x,t}$ was calculated using the following formula:

$$AF_{x,t} = 1 - e^{(-\sum_{l=l_0}^L \beta_{x_{t-l}})}$$

where the parameter β_x represents the risk associated with exposure x relative to a reference value of x_0 (in our case the Minimum Mortality Temperature) and corresponds to logarithm of the odds ratio; l_0 and L represent the lowest and the highest number of lags considered, respectively; and $\sum_{l=l_0}^L \beta_{x_{t-l}}$ represents the overall cumulative exposure-response association, which is composed of the sum of contributions β_{x_l} from exposures $x_{t-l_0}, \dots, x_{t-L}$ experienced within the lag period. Thus, $AF_{x,t}$ can be interpreted as the fraction of neonatal mortality at time t attributable to past temperature exposures in the period $t - l_0, \dots, t - L$ compared to a constant exposure x_0 throughout the same period. The attributable fractions were separated into components related to hot and cold temperatures by restricting the analysis to temperature ranges above and below the reference temperature percentiles for neonatal mortality. These attributable risk fractions can be interpreted as individual's probability of dying and summed up to derive attributable fraction at the sampled population level. We subsequently obtained the excess fraction of neonatal deaths attributable to climate change (AF_{CC}) by subtracting the hot- and cold-related burdens between the factual and counterfactual scenarios. We computed the climate change-attributable heat- and cold-related neonatal mortality rates per 100,000 live births for each country as the climate attributable fractions multiplied by the respective neonatal mortality rates

$$AR_{neonatal\ mortality} = m_{neonatal} \cdot AF_{CC}$$

where $m_{neonatal}$ refers to the neonatal mortality rate and AF_{CC} – to the fraction of heat- or cold-related neonatal mortality attributable to climate change. We have now added these additional explanations in the Methods section.

Fu, S. H., Gasparrini, A., Rodriguez, P. S., & Jha, P. (2018). Mortality attributable to hot and cold ambient temperatures in India : a nationally representative case-crossover study. *PLoS Med*, 15(7), 1–17.

Gasparrini, A., & Leone, M. (2014). Attributable risk from distributed lag models. *BMC Medical Research Methodology*, 14(1), 1–8. <https://doi.org/10.1186/1471-2288-14-55>

He, Q., Liu, Y., Yin, P., Gao, Y., Kan, H., Zhou, M., Chen, R., & Li, Y. (2023). Differentiating the impacts of ambient temperature on pneumonia mortality of various infectious causes: a nationwide, individual-level, case-crossover study. *EBioMedicine*, 98, 104854. <https://doi.org/10.1016/j.ebiom.2023.104854>

Maclure, M. (1991). The Case-Crossover Design: A Method for Studying Transient Effects on the Risk of Acute Events. *Am J Epidemiol*, 133(2), 144–153. [internal-pdf://244.176.220.64/Am. J. Epidemiol.-1991-Maclure-144-53.pdf](https://doi.org/10.1093/oxfordjournals.aje.a111111)

Navidi, W. (1998). Bidirectional Case-Crossover Designs for Exposures with Time Trends. *Biometrics*, 54(2), 596. <https://doi.org/10.2307/3109766>

2. The authors have mentioned “up to seven days of lag” that were used in modelling choice. But it seems that only the result of AIC on lag 0-2 days is provided.

Response: We actually used the AIC criterion only to guide the choice of the number and placement of knots in our models. The choice of maximum lag was based on *a priori* visual examination of the temperature-mortality relationship for specific temperatures and lags. This is a standard approach applied in studies using the DLNM framework (Gasparrinia et al., 2010). The plots of the temperature-mortality relationship for specific temperatures and lags were provided in the previous round of review and we have now also included them in the Supplementary Material (Supplementary Fig. 3 and Supplementary Fig. 4).

Gasparrinia, A., Armstrong, B., & Kenward, M. G. (2010). Distributed lag non-linear models. *Statistics in Medicine*, 29(21), 2224–2234. <https://doi.org/10.1002/sim.3940>

3. Considering the huge heterogeneity of risk of temperature across the country, when calculating the total effect across country, it is better to include the country variable as a random-effect variable in the model.

We actually consider differences in temperature-related mortality risk within a country by standardising absolute temperatures in DHS cluster-specific temperature percentiles. This allows for the minimum mortality temperature and risks to vary across different locations in the country. As clarified in the discussion, this approach allows to account for population adaptation to their predominant climate to some extent, although the shape of the exposure-response functions across locations is assumed to be the same in a percentile scale.

Thank you for the suggestion of including a random effect for country. The available frequentist software for conditional logistic regression does not easily allow incorporating a random effect for country. Barrera-Gómez et al. (2023) recently introduced a Bayesian estimation procedure for this purpose, but their method is based on a conditional Poisson regression and not a logistic regression, so it cannot be easily applied in our case. Furthermore, Barrera-Gómez et al. (2023) indicated that the one-stage analysis with random effects produced similar results to the standard two-stage approach

that we already tested. As explained in the previous round of review, the two-stage analysis yielded similar point estimates to the one-stage analysis that we use despite the wider CIs. We have also already acknowledged in the limitations of our analyses that we cannot obtain country-specific differences in risk due to the limited statistical power.

Barrera-Gómez, J., Puig, X., Ginebra, J., & Basagaña, X. (2023). Conditional Poisson Regression with Random Effects for the Analysis of Multi-site Time Series Studies. *Epidemiology*, 34(6), 873–878. <https://doi.org/10.1097/EDE.0000000000001664>

4. In comparison with the temperature percentile, absolute temperature is more familiar to the public and policymakers. Therefore, I will suggest some key results using absolute temperature.

It is true that absolute temperatures are more often used and thus more familiar to the public and policymakers. However, there is evidence across many outcomes that population acclimatise to their local temperature, and a percentile scale is able to account for this issue.

For the purpose of better communicating the results to the public and policymakers, we have now produced a map, which shows geographic variations (based on DHS cluster) of the Minimum Mortality Temperature (MMT) in absolute temperature (See Extended Data Fig. 1). We also included tables with the mean, minimum and maximum of the MMT observed across different DHS clusters within a country for the study period (See Supplementary Table 2 and 3).

Minor comment:

1. The “relative risk (RR)” reported in the method and result section should be the “odds ratio (OR)”.

Since we apply a logistic regression in a time-stratified case-crossover design the coefficients derived from the regression analysis represent (log) RR, not OR (Labrecque et al., 2021; Rothman et al., 2008).

Labrecque, J. A., Hunink, M. M. G., Ikram, M. A., & Ikram, M. K. (2021). Do Case-Control Studies Always Estimate Odds Ratios? *American Journal of Epidemiology*, 190(2), 318–321. <https://doi.org/10.1093/aje/kwaa167>

Rothman, K., Greenland, S., & Lash, T. (2008). Chapter 8: Case-Control Studies. In *Modern Epidemiology* (3rd ed.). Lippincott Williams & Wilkins.

2. For the period of temperature-related burden estimation, “2000-2019” is more common in previous studies than “2001-2019”. In addition, for the cutoffs to define moderate cold and heat, I will suggest using “25th” and “75th” rather than “25.5th” and “75.5th”.

Response: We decided to include the period 2001-2019 as the exposure-response functions we developed were based on neonatal mortality data from this time period and we did not want to extrapolate them beyond. Although done in certain studies, extrapolating exposure-response functions over periods where data are not available requires stronger assumptions.

Thank you for the suggestion, we have now changed the cut-offs to “25th” and “75th”.

3. Line 166, this subtitle should be “Temperature-related neonatal deaths attributable to climate change”

Response: Thank you noticing this, we have now corrected the subtitle accordingly.

4. Line 380, surveys were collected only between 2014 and 2021, which seems to conflict periods (5 and 10 years prior to the interview) mentioned in Line 461.

It is true that the included DHS surveys were collected between 2014 and 2021. However, the data collection period for each country was different, so that the interview date was not consistent across surveys. Please note that year of survey collection refers to when the survey was conducted and not the year of birth/death of each child. Information on the year of birth/death of each child was collected retrospectively by asking mothers about their birth histories. In the final analysis we restricted the neonatal deaths to those reported by the mother within 15 years prior to the interview date. We conducted a sensitivity analysis to test for potential problems with misreporting or inaccurate reporting of deaths by restricting neonatal deaths to those reported by the mother within the last 5 and 10 years prior to the interview date.

5. Lines 458-459, please specify the information on the number and placement of knots for exposure-response function.

Response: Thank you for this suggestion. We have now included reference to Supplementary Table 6, which provides an overview of the different model configurations tested based on number and placement of knots.

6. Correlation test is required for the Extended data Figure 1.

Response: Thank you for the suggestion. We have now included Pearson correlation test results to Extended data Figure 1. Please note that the test has been added only for the plots with mean ambient temperature and MMT. For average latitude and MMT correlation test is not meaningful as the relationship is non-linear.

Extended Data Fig. 1: Annual mean temperature and minimum mortality temperature in the 29 countries. a, Annual mean temperature and minimum neonatal mortality temperature (°C). **b,** Average latitude and minimum neonatal mortality temperature (°C) **c,** Annual mean temperature and minimum very early neonatal mortality temperature (°C). **d,** Average latitude and minimum very early neonatal mortality temperature (°C)

Reviewer #1 (Remarks on code availability):

The authors have not provided the code for reviewing.

Response: We apologise for not providing the code earlier. It has been now uploaded on Code Ocean.

Reviewer #2 (Remarks to the Author):

The authors have sufficiently addressed my comments.

Reviewer #3 (Remarks to the Author):

The authors have adequately addressed my comments and suggestions. Thank you for the work on improving the manuscript. I recommend the manuscript for publication.

Reviewer #3 (Remarks on code availability):

There is no code available to review.

Response: We apologise for not providing the code earlier. It has been now uploaded on Code Ocean.